# So3krates: Equivariant attention for interactions on arbitrary length-scales in molecular systems

**J. Thorben Frank**[1,2]*  **Oliver T. Unke**[1,2,3]

**Klaus-Robert Müller**[1,2,3,4,5]†

[1] Machine Learning Group, TU Berlin, 10587 Berlin, Germany
[2] BIFOLD, Berlin Institute for the Foundations of Learning and Data, Germany
[3] Google Research, Brain team, Berlin
[4] Department of Artificial Intelligence, Korea University, Seoul 136-713, Korea
[5] Max Planck Institut für Informatik, 66123 Saarbrücken, Germany

## Abstract

The application of machine learning methods in quantum chemistry has enabled the study of numerous chemical phenomena, which are computationally intractable with traditional *ab-initio* methods. However, some quantum mechanical properties of molecules and materials depend on non-local electronic effects, which are often neglected due to the difficulty of modeling them efficiently. This work proposes a modified attention mechanism adapted to the underlying physics, which allows to recover the relevant non-local effects. Namely, we introduce *spherical harmonic coordinates* (SPHCs) to reflect higher-order geometric information for each atom in a molecule, enabling a non-local formulation of attention in the SPHC space. Our proposed model SO3KRATES[3] – a self-attention based message passing neural network – uncouples geometric information from atomic features, making them independently amenable to attention mechanisms. Thereby we construct *spherical filters*, which extend the concept of continuous filters in Euclidean space to SPHC space and serve as foundation for a *spherical self-attention* mechanism. We show that in contrast to other published methods, SO3KRATES is able to describe non-local quantum mechanical effects over arbitrary length scales. Further, we find evidence that the inclusion of higher-order geometric correlations increases data efficiency and improves generalization. SO3KRATES matches or exceeds state-of-the-art performance on popular benchmarks, notably, requiring a significantly lower number of parameters (0.25–0.4x) while at the same time giving a substantial speedup (6–14x for training and 2–11x for inference) compared to other models.

## 1 Introduction

Atomistic simulations use long time-scale molecular dynamics (MD) trajectories to predict macroscopic properties that arise from interactions on the microscopic scale [1–3]. Their predictive reliability is determined by the accuracy of the underlying *force field* (FF), which needs to be queried at every time step. This quickly becomes a computational bottleneck if the forces are determined from first principles, which may be required for accurate results. To that end, machine learning

---

*thorbenjan.frank@googlemail.com
†klaus-robert.mueller@tu-berlin.de

[3] https://github.com/thorben-frank/mlff

36th Conference on Neural Information Processing Systems (NeurIPS 2022).

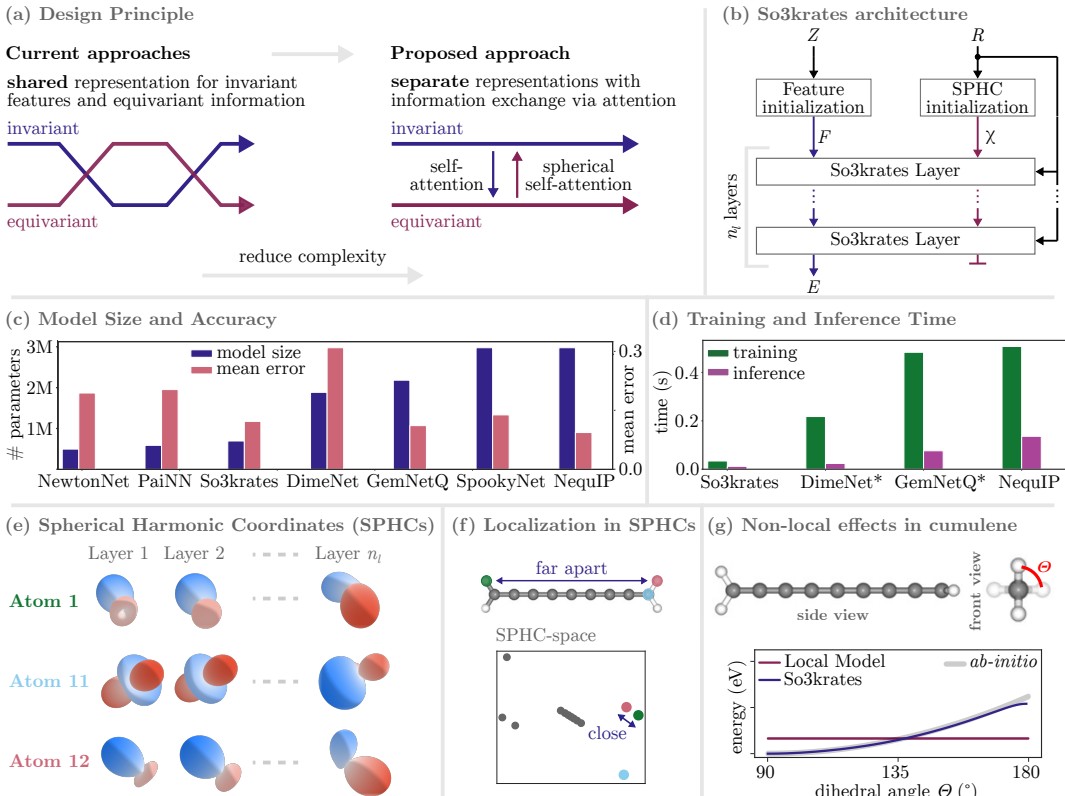

Figure 1: **(a)** Design principle of our proposed message-passing scheme, where invariant features **f** and equivariant information - represented by *spherical harmonic coordinates* (SPHCs) $\chi$ - are separate and exchange information via (spherical) self-attention. **(b)** Overview of the SO3KRATES architecture. **(c)** Comparison of prediction accuracy and model size of different architectures. **(d)** Comparison of training and inference times of different architectures. **(e)** Illustration of learned SPHCs at different layers of the SO3KRATES architecture. **(f)** Low-dimensional projection of atomic SPHCs, showing that atoms far apart in Euclidean space can be mapped close together in SPHC space. **(g)** In contrast to local models, SO3KRATES is able to learn the long-range correlations between hydrogen rotors in cumulene and reproduces the *ab-initio* ground truth faithfully.

FFs (MLFFs) offer a computationally more efficient, yet accurate empirical alternative to expensive *ab-initio* methods [2, 4–24].

In recent years, *Geometric Deep Learning* has become a popular design paradigm, which exploits relevant symmetry groups of the underlying learning problem by incorporating a *geometric prior* [12, 25, 26]. This effectively restricts the learnable functions of the model to a subspace with a meaningful inductive bias. Prominent examples for such models are e.g. convolutional neural networks (CNNs) [27], which are equivariant w.r.t. the group of translations, or graph neural networks (GNNs) [28], which are invariant w.r.t. node permutation.

For molecular property prediction, it has been shown that equivariance w.r.t. the 3D rotation group $SO(3)$ greatly improves data efficiency and accuracy of the learned FFs [29–32]. To achieve equivariance, architectures either rely on feature expansions in terms of spherical harmonics (SH) [33] or explicitly include (dihedral) angles [29, 34]. While the latter scales quadratically (cubically) in the number of neighboring atoms and has been shown to be geometrically incomplete [35], the calculation of spherical harmonics scales only linear in the number of neighboring atoms, which makes them a fast and accurate alternative [30, 32, 36].

---

[*] Reference times were taken from [34]. As our own timings were measured on a different GPU, we decreased the reported times according to speedup-factors reported in [37]. For full details, see appendix A.6.

However, current higher-order geometric representations based on SHs usually result in expensive transformations, since an individual feature channel per SH degree (and order) is required. As a result, going to higher degrees is computationally expensive and comes at the price of increasing complexity, resulting in state-of-the-art (SOTA) models with millions of parameters [30, 32, 34]. However, in order to be applicable to large molecular structures, models are required to be both efficient and accurate on *all* length scales.

Non-local electronic effects have been outlined as one of the major challenges for a new generation of MLFFs [21]. They result in non-local, higher-order geometric relations between atoms. Most current architectures implicitly assume locality of interactions (expressed through a local neighborhood), which prohibits an efficient description of all relevant atomic interactions at larger scales. Simply increasing the cutoff radius used to determine local neighborhoods is not an adequate solution, since it only shifts the problem to larger length scales [30].

In this work, we propose *spherical harmonic coordinates* (SPHCs), which encode higher-order geometric information for each node in a molecular graph (Fig. 1e). This is in stark contrast to current approaches, which consider molecules as three-dimensional point clouds with learned features and fixed atomic coordinates: we propose to make the SPHCs themselves a *learned* quantity. Through localization in the space of SPHCs (Fig. 1f), models are able to efficiently describe electronic effects that are non-local in three-dimensional Euclidean space (Fig. 1g).

We then present SO3KRATES (Fig. 1b), a self-attention based message-passing neural network (MPNN), which decouples atomic features and SPHCs and updates them individually (Fig. 1a). This resembles ideas from equivariant graph neural networks [25], but allows to go to arbitrarily high geometric orders. The separation of higher-order geometric and feature information allows to overcome the parametric and computational complexity usually encountered in models with higher-order geometric representations, since we only require a single feature channel (instead of one per SH degree and order). Thus, SO3KRATES resembles some early architectures like SCHNET [10] or PHYSNET [19] in parametric simplicity. We further show that SO3KRATES outperforms the popular sGDML [38] kernel model by a large margin in the low-data regime, a domain which has so far been considered to be dominated by kernel machines [21]. Numerical evidence suggests that the data efficiency of SO3KRATES is directly related to the maximal degree of geometric information encoded in the SPHCs. We then apply SO3KRATES to the well-established MD17 benchmark and show that our model achieves SOTA results, despite is light-weight structure and having only 0.25–0.4x the number of parameters of competitive architectures (Fig. 1c), while achieving speedups of 6–14x and 2–11x for training and inference, respectively (Fig. 1d).

Although we focus on quantum chemistry applications in this work, the developed methods are also applicable to other fields where long-ranged correlations in three-dimensional data are relevant. For example, models based on SPHCs may also be applicable to tasks like 3D shape classification or computer vision.

## 2    Preliminaries and Related Work

In the following, we review the most important concepts our method is based upon and relate it to prior work.

**Message Passing Neural Networks**    MPNNs [14] carry over many of the benefits of convolutions to unstructured domains and have thus become one of the most promising approaches for the description of molecular properties. Their general working principle relies on the repeated iteration of message passing (MP) steps, which can be phrased as follows [14, 25]

$$\mathbf{m}_{ij} = m(\mathbf{f}_i, \mathbf{f}_j, \boldsymbol{r}_{ij}) \tag{1}$$

$$\mathbf{m}_i = \sum_{j \in \mathcal{N}(i)} \mathbf{m}_{ij} \tag{2}$$

$$\mathbf{f}'_i = u(\mathbf{f}_i, \mathbf{m}_i). \tag{3}$$

Here, $\mathbf{m}_{ij}$ is the message between atoms $i$ and $j$ computed with the message function $m(\cdot)$, $\mathbf{m}_i$ is the aggregation of all messages in the neighborhood $\mathcal{N}(i)$ of atom $i$, and $u(\cdot)$ is an update function returning updated features $\mathbf{f}'_i$ based on the current features $\mathbf{f}_i$ and message $\mathbf{m}_i$. The neighborhood

$\mathcal{N}(i)$ consists of all atoms which lie within a given cutoff radius around the atomic position $\boldsymbol{r}_i$, which ensures linear scaling in the number of atoms $n$. While earlier variants parametrized messages only in terms of inter-atomic distances [13, 19], more recent approaches also take higher-order geometric information into account [25, 29, 31, 32, 39, 40].

**Molecules as Point Clouds**  A molecule can be considered as a point cloud of $n$ atoms $\mathcal{P}_{3D}(\mathcal{R}, \mathcal{F})$, where $\mathcal{R} = (\boldsymbol{r}_1, ... \boldsymbol{r}_n)$ denotes the set of atomic positions $\boldsymbol{r}_i \in \mathbb{R}^3$ and $\mathcal{F} = (\mathbf{f}_1, \ldots, \mathbf{f}_n)$ is the set of rotationally invariant atomic descriptors, or features, $\mathbf{f}_i \in \mathbb{R}^F$. We write the distance vector pointing from the position of atom $i$ to the position of atom $j$ as $\boldsymbol{r}_{ij} = \boldsymbol{r}_j - \boldsymbol{r}_i$, the distance as $r_{ij} = \|\boldsymbol{r}_{ij}\|_2$ and the normalized distance vector as $\hat{\boldsymbol{r}}_{ij} = \boldsymbol{r}_{ij}/r_{ij}$. Given the point cloud, a density over Euclidean space assigning a vector value to each point $\boldsymbol{r}$ can be constructed as

$$\boldsymbol{\rho}(\boldsymbol{r}) = \sum_{i=1}^{n} \delta(\|\boldsymbol{r}_i - \boldsymbol{r}\|_2) \cdot \mathbf{f}_i \,, \tag{4}$$

where $\delta$ is the Dirac delta function. It can be shown that applying a convolutional filter on $\boldsymbol{\rho}(\boldsymbol{r})$ resembles the update steps used in MPNNs [41].

**Equivariance**  Given a set of transformations that act on a vector space $\mathbb{A}$ as $S_g : \mathbb{A} \mapsto \mathbb{A}$ to which we associate an abstract group $G$, a function $f : \mathbb{A} \mapsto \mathbb{B}$ is said to be equivariant w.r.t. $G$ if

$$f(S_g x) = T_g f(x) \,, \tag{5}$$

where $T_g : \mathbb{B} \mapsto \mathbb{B}$ is an equivalent transformation on the output space [25]. Thus, in order to say that $f$ is equivariant, it must hold that under transformation of the input, the output transforms "in the same way". While equivariance has been a popular concept in signal processing for decades (cf. e.g. [42] or wavelet neural networks [43]), recent years have seen efforts to design group equivariant NNs and kernel methods, since respecting relevant symmetries builds an important inductive bias [12, 44, 45]. Examples are CNNs [27] which are equivariant w.r.t. translation, GNNs [14, 28] which are invariant ($T_g = \mathbb{I}$) w.r.t. permutation, or architectures which are equivariant w.r.t. the $SO(3)$ group [25, 31, 33, 36, 46]. In this work, we consider the $SO(3)$ group of rotations, such that $\mathbb{A}$ is the Euclidean space $\mathbb{R}^3$, where the corresponding group actions are given by rotation matrices $R_\theta \in \mathbb{R}^{3 \times 3}$.

**Spherical Harmonics**  The spherical harmonics are special functions defined on the surface of the sphere $S^2 = \{\hat{\boldsymbol{r}} \in \mathbb{R}^3 : \|\hat{\boldsymbol{r}}\|_2 = 1\}$ and form an orthonormal basis for the irreducible representations (irreps) of $SO(3)$. In the context of *tensor field networks* [33], they have been introduced as elementary building blocks for $SO(3)$-equivariant neural networks. The spherical harmonics are commonly denoted as $Y_l^m(\hat{\boldsymbol{r}}) : S^2 \mapsto \mathbb{R}$, where the *degree* $l$ determines all possible values of the *order* $m \in \{-l, \ldots, +l\}$. They transform under rotation as

$$Y_l^m(R_\theta \, \hat{\boldsymbol{r}}) = \sum_{m'} D_{mm'}^l(R_\theta) Y_l^{m'}(\hat{\boldsymbol{r}}) \,, \tag{6}$$

where $D_{mm'}^l(R_\theta)$ are the entries of the *Wigner-D* matrix $\mathbf{D}^l(R_\theta) \in \mathbb{R}^{(2l+1) \times (2l+1)}$ [47]. Based on the spherical harmonics, we define a vector-valued function $\boldsymbol{Y}^{(l)} : S^2 \mapsto \mathbb{R}^{2l+1}$ for each degree $l$, with entries $Y_l^m$ for all valid orders $m$ of a given degree $l$. Since $\boldsymbol{Y}^{(l)}(R_\theta \, \hat{\boldsymbol{r}}) = \mathbf{D}^l(R_\theta)\boldsymbol{Y}^{(l)}(\hat{\boldsymbol{r}})$ (cf. eq. (6)), $\boldsymbol{Y}^{(l)}$ is equivariant w.r.t. $SO(3)$.

**Tensor Product Contractions**  The irreps $\boldsymbol{Y}^{(l_1)}$ and $\boldsymbol{Y}^{(l_2)}$ can be coupled by computing their tensor product $\boldsymbol{Y}^{(l_1)} \otimes \boldsymbol{Y}^{(l_2)}$, which can equivalently be expressed as a direct sum [33, 48]

$$\boldsymbol{Y}^{(l_1)} \otimes \boldsymbol{Y}^{(l_2)} = \bigoplus_{l_3=|l_1-l_2|}^{l_1+l_2} \overbrace{\widetilde{\boldsymbol{Y}}^{(l_3)}}^{:=\left(\boldsymbol{Y}^{(l_1)} \otimes_{l_3} \boldsymbol{Y}^{(l_2)}\right)} \,, \tag{7}$$

where the entry of order $m_3$ for the coupled irreps $\widetilde{\boldsymbol{Y}}^{(l_3)}$ is given by

$$\tilde{Y}_{m_3}^{l_3} = \sum_{m_1=-l_1}^{l_1} \sum_{m_2=-l_2}^{l_2} C_{m_1,m_2,m_3}^{l_1,l_2,l_3} Y_{m_1}^{l_1} Y_{m_2}^{l_2} \,, \tag{8}$$

and $C_{m_1,m_2,m_3}^{l_1,l_2,l_3}$ are the so-called *Clebsch-Gordon coefficients*. In the following, we will denote the tensor product of degrees $l_1$ and $l_2$ followed by "contraction" to $l_3$ (meaning the irreps of degree $l_3$ in the direct sum representation of their tensor product) as $\left(\boldsymbol{Y}^{(l_1)} \otimes_{l_3} \boldsymbol{Y}^{(l_2)}\right)$, which is a mapping of the form $\mathbb{R}^{(2l_1+1)\times(2l_2+1)} \mapsto \mathbb{R}^{2l_3+1}$, since $m_3 \in \{-l_3, \dots l_3\}$.

## 3 Methods

In the following, we describe the main methodological contributions of this work. We introduce the concept of an adapted point cloud $\mathcal{P}_{3D}(\mathcal{R}, \mathcal{X}, \mathcal{F})$, which incorporates the set of spherical harmonics coordinates (SPHCs) $\mathcal{X} = (\boldsymbol{\chi}_1, \dots, \boldsymbol{\chi}_n)$ (see below) in addition to features $\mathcal{F}$ and Euclidean coordinates $\mathcal{R}$. However, contrary to $\mathcal{R}$, SPHCs $\mathcal{X}$ are refined during the message passing updates. Having SPHCs as part of the molecular point cloud extends the idea of current MPNNs, which learn message functions on $\mathcal{R}$, only. Instead, we learn a message function $m$ (cf. eq. (1)) on both, the (fixed) atomic coordinates $\mathcal{R}$ as well as on the SPHCs $\mathcal{X}$. This adapted message-passing scheme allows to learn non-local geometric corrections. Based on these design principles, we propose the SO3KRATES architecture.

**Initialization**  Feature vectors are initialized from the atomic numbers $z_i \in \mathbb{N}$ (denoting which chemical element an atom belongs to) by an embedding map

$$\mathbf{f}_i = f_{\mathrm{emb}}(z_i), \tag{9}$$

where $f_{\mathrm{emb}} : \mathbb{N} \mapsto \mathbb{R}^F$. We define SPHCs $\boldsymbol{\chi}$ as the concatenation of degrees $\mathcal{L} := \{l_{\min}, \dots, l_{\max}\}$

$$\boldsymbol{\chi} = [\ \underbrace{\boldsymbol{\chi}^{(l_{\min})}}_{\in \mathbb{R}^{2l_{\min}+1}}, \dots, \underbrace{\boldsymbol{\chi}^{(l_{\max})}}_{\in \mathbb{R}^{2l_{\max}+1}}\ ] \in \mathbb{R}^{(l_{\max}-l_{\min}+1)^2}, \tag{10}$$

such that their transformation under rotation can be expressed in terms of concatenated Wigner-D matrices (see appendix A.1). The short-hand $\boldsymbol{\chi}^{(l)} \in \mathbb{R}^{2l+1}$ refers to the subset of SPHCs with degree $l$. They are initialized as

$$\boldsymbol{\chi}_i^{(l)} = \frac{1}{\mathcal{C}_i} \sum_{j \in \mathcal{N}(i)} \phi_{r_{\mathrm{cut}}}(r_{ij}) \cdot \boldsymbol{Y}^{(l)}(\hat{\boldsymbol{r}}_{ij}), \tag{11}$$

where $\mathcal{C}_i = \sum_{j \in \mathcal{N}(i)} \phi_{r_{\mathrm{cut}}}(r_{ij})$, $\phi_{r_{\mathrm{cut}}} : \mathbb{R} \mapsto \mathbb{R}$ is the cosine cutoff function [6], and the sum runs over the neighborhood $\mathcal{N}(i)$ of atom $i$.

**Message Passing Update**  Two branches of attention-weighted MP steps are defined for the feature vectors $\mathbf{f}$ and SPHCs $\boldsymbol{\chi}$ (see Fig. 1a). After initialization (eqs. (9) and (11)), the features are updated as

$$\mathbf{f}_i' = \mathbf{f}_i + \sum_{j \in \mathcal{N}(i)} \phi_{r_{\mathrm{cut}}}(r_{ij}) \cdot \alpha_{ij} \cdot \mathbf{f}_j, \tag{12}$$

where $\alpha_{ij} \in \mathbb{R}$ are self-attention [49, 50] coefficients (see below). In analogy to the feature vectors, it is possible to define an MP update for the SPHCs as

$$\boldsymbol{\chi}_i'^{(l)} = \boldsymbol{\chi}_i^{(l)} + \sum_{j \in \mathcal{N}(i)} \phi_{r_{\mathrm{cut}}}(r_{ij}) \cdot \alpha_{ij}^{(l)} \cdot \boldsymbol{Y}^{(l)}(\hat{\boldsymbol{r}}_{ij}), \tag{13}$$

where individual attention coefficients $\alpha_{ij}^{(l)} \in \mathbb{R}$ for each degree of the SPHCs are computed using multi-head attention [49]. However, with this definition, both MP updates are limited to local neighborhoods $\mathcal{N}(i)$. To be able to model non-local effects, we introduce the SPHC distance matrix $\mathbf{X} \in \mathbb{R}^{n \times n}$ with entries $\chi_{ij} = \|\boldsymbol{\chi}_i - \boldsymbol{\chi}_j\|_2$, i.e. distances between two atoms $i$ and $j$ in SPHC space for all possible pair-wise combinations of $n$ atoms. To have uniform scales, we further apply the softmax along each row of $\mathbf{X}$ to generate a rescaled matrix $\tilde{\mathbf{X}} = \mathrm{softmax}(\mathbf{X})$ with entries $\tilde{\chi}_{ij}$. A polynomial cutoff function $\phi_{\tilde{\chi}_{\mathrm{cut}}}$ [29] is then applied to $\tilde{\mathbf{X}}$ to define spherical neighborhoods $\mathcal{N}_{\boldsymbol{\chi}}(i)$ (see A.2), which may include atoms that are far away in Euclidean space (see Fig. 1f). The spherical cutoff distance is chosen as $\tilde{\chi}_{\mathrm{cut}} = 1/n$ to ensure that spherical neighborhoods remain small, even

when going to larger molecules. We then incorporate non-local geometric corrections into the MP update of the SPHCs as

$$\boldsymbol{\chi}_i'^{(l)} = \boldsymbol{\chi}_i^{(l)} + \underbrace{\sum_{j \in \mathcal{N}(i)} \phi_{r_{\mathrm{cut}}}(r_{ij}) \cdot \alpha_{ij}^{(l)} \cdot \boldsymbol{Y}^{(l)}(\hat{\boldsymbol{r}}_{ij})}_{\text{local in } \mathbb{R}^3} + \underbrace{\sum_{j \in \mathcal{N}_{\boldsymbol{\chi}}(i)} \phi_{\tilde{\chi}_{\mathrm{cut}}}(\tilde{\chi}_{ij}) \cdot \alpha_{ij}^{(l)} \cdot \boldsymbol{Y}^{(l)}(\hat{\boldsymbol{r}}_{ij})}_{\text{local in } \boldsymbol{\chi}, \text{ but non-local in } \mathbb{R}^3} . \quad (14)$$

We will show in the first part of the experiments, how geometric corrections from SPHC space allow for modelling non-local quantum effects, inaccessible to current architectures. In the second part, we use a SO3KRATES model without geometric corrections, which makes it a traditional MPNN in the sense of only localizing in $\mathbb{R}^3$. We find this architecture to be highly parameter, data and time efficient while capable of reaching SOTA results.

**Spherical Filter and Self-Attention** The self-attention coefficients in eqs. (12)–(14) are calculated as

$$\alpha_{ij} = \mathbf{f}_i^T (\mathbf{w}_{ij} \odot \mathbf{f}_j) / \sqrt{F} , \quad (15)$$

where $\mathbf{w}_{ij} \in \mathbb{R}^F$ is the output of a filter generating function and '$\odot$' denotes the element-wise product. The filter maps the Euclidean distance $r_{ij}$ and per-degree SPHC distances $\chi_{ij}^{(l)} = \|\boldsymbol{\chi}_j^{(l)} - \boldsymbol{\chi}_i^{(l)}\|_2$ between the current SPHCs of atoms $i$ and $j$ into the feature space $\mathbb{R}^F$ (as a short-hand, we write the vector containing all per-degree SPHC distances as $[\chi_{ij}^{(l)}]_{l \in \mathcal{L}}$). It is built as the linear combination of two filter-generating functions

$$\mathbf{w}_{ij} = \underbrace{\phi_r(r_{ij})}_{\text{radial filter}} + \underbrace{\phi_s\left([\chi_{ij}^{(l)}]_{l \in \mathcal{L}}\right)}_{\text{spherical filter}}, \quad (16)$$

which separately act on the Euclidean and SPHC distances. We call $\phi_r : \mathbb{R} \mapsto \mathbb{R}^F$ the *radial filter* function and $\phi_s : \mathbb{R}^{|\mathcal{L}|} \mapsto \mathbb{R}^F$ the *spherical filter* function (an ablation study for $\phi_s$ can be found in appendix A.4). Since per-atom features $\mathbf{f}_i$, interatomic distances $r_{ij}$, and per-degree distances $\chi_{ij}^{(l)}$ are invariant under rotations (proof in appendix A.1), so are the self-attention coefficients $\alpha_{ij}$.

While we choose to pass the per-degree norms directly into the filter generating function $\phi_s$, future work might explore the possibilities of alternative metrics (instead of the L2 norm) or an expansion in terms of basis functions as it is common practice for inter-atomic distances (see appendix A.3 eq. (30)).

**Atomwise Interaction** After each MP update, features and SPHCs are coupled with each other according to

$$\mathbf{f}_i' = \mathbf{f}_i + \phi_1\left(\mathbf{f}_i, [\chi_i^{(l)}]_{l \in \mathcal{L}}, [\tilde{\chi}_i^{(l)}]_{l \in \mathcal{L}}\right) , \quad (17)$$

$$\boldsymbol{\chi}_i'^{(l)} = \boldsymbol{\chi}_i^{(l)} + \phi_2^{(l)}\left(\mathbf{f}_i, [\chi_i^{(l)}]_{l \in \mathcal{L}}, [\tilde{\chi}_i^{(l)}]_{l \in \mathcal{L}}\right) \boldsymbol{\chi}_i^{(l)} + \phi_3^{(l)}\left([\tilde{\chi}_i^{(l)}]_{l \in \mathcal{L}}\right) \tilde{\boldsymbol{\chi}}_i^{(l)} , \quad (18)$$

where $\phi_1 : \mathbb{R}^{F+2|\mathcal{L}|} \mapsto \mathbb{R}^F$, $\phi_2^{(l)} : \mathbb{R}^{F+2|\mathcal{L}|} \mapsto \mathbb{R}$, and $\phi_3^{(l)} : \mathbb{R}^{|\mathcal{L}|} \mapsto \mathbb{R}$. In the inputs to $\phi_{1,2,3}$, degree-wise scalars $\chi^{(l)} = \|\boldsymbol{\chi}^{(l)}\|_2$ are used to preserve equivariance. The coupling step additionally includes cross-degree coupled SPHCs $\tilde{\boldsymbol{\chi}}_i^{(l)}$ for each degree $l$. Following [48] they are constructed as

$$\tilde{\boldsymbol{\chi}}_i^{(l)} = \sum_{l_1 = l_{\min}}^{l_{\max}} \sum_{l_2 = l_1 + 1}^{l_{\max}} k_{l_1, l_2, l} \left(\boldsymbol{\chi}_i^{(l_1)} \otimes_l \boldsymbol{\chi}_i^{(l_2)}\right), \quad (19)$$

where $k_{l_1, l_2, l} \in \mathbb{R}$ are learnable coefficients for all valid combinations of $l_1, l_2$ given $l$ and the term in brackets is the contraction of degrees $l_1$ and $l_2$ into degree $l$ (eq. (8)).

**SO3KRATES architecture** Using the design paradigm above, we build the transformer network SO3KRATES, which consists of a self-attention block on $\mathcal{F}$ and $\mathcal{X}$ (eqs. (12) and (13)), respectively, as well as an interaction block (eqs. (17) and (18)) per layer. After initialization of the features and the SPHCs according to eqs. (9) and (11), they are updated iteratively by passing through $n_l$ layers. Atomic energy contributions $E_i \in \mathbb{R}$ are predicted from the features of the final layer using a two-layered output block. The individual contributions are summed to the total energy prediction $E = \sum_i^n E_i$. See Fig. 1b for an overview. More details on the implementation, training details and network hyperparameters are given in appendix A.3 and A.13.

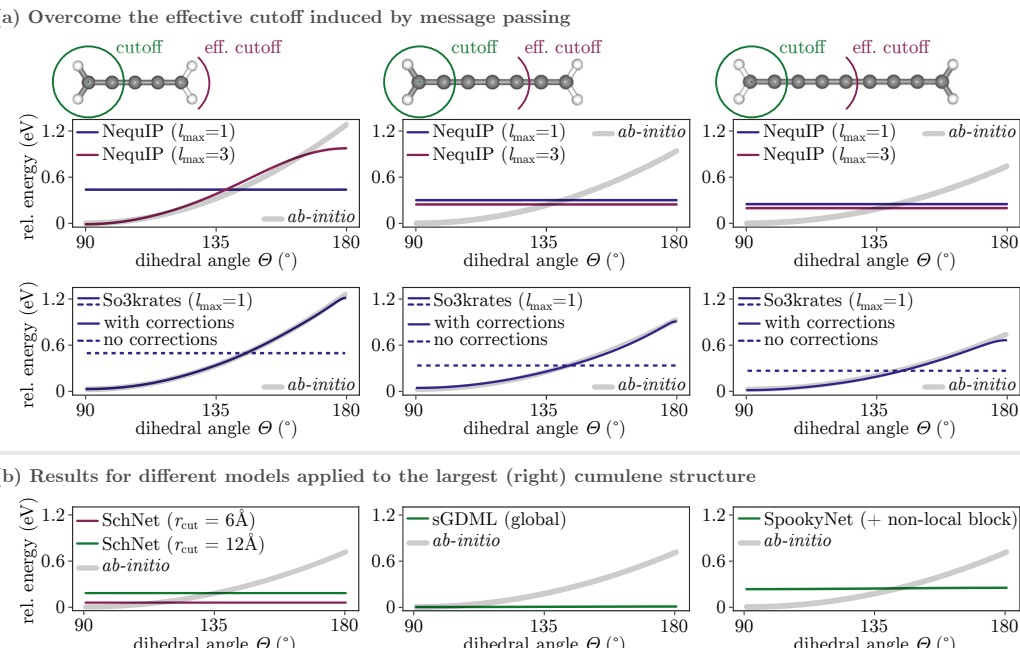

Figure 2: **(a)** Energy predictions for different cumulene structures with NEQUIP ($l_{\max} = 1$ and $l_{\max} = 3$) and SO3KRATES ($l_{\max} = 1$) models with and without geometric corrections (eqs. (13) and (14)). Cutoff radius and number of layers are kept constant at $r_{\text{cut}} = 2.5\,\text{Å}$ and $n_l = 4$, respectively. **(b)** We additionally test other SOTA models on the largest cumulene structure (9 carbons). We find, that even for a very large cutoff, SCHNET fails to describe the dihedral angle profile. The same holds true for an inherently global sGDML model and even for a SPOOKYNET model that explicitly includes non-local corrections.

## 4  Experiments

In the first subsection, we show how non-local quantum effects can be incorporated by using non-local corrections from the space of SPHCs. In the second part of the experiments, we remove the non-local part which yields a traditional, $\mathbb{R}^3$-local MPNN which reaches SOTA results on established benchmarks while requiring much less computational time and parameters than competitive models. A scaling analysis as well as an accuracy comparison for both model variants can be found in appendix A.8 and A.5.

**Non-Local Geometric Interactions**   For efficiency reasons, MPNNs only consider interactions between atoms in local neighborhoods, i.e. within a cutoff radius $r_{\text{cut}}$. Thus, information can only be propagated over a distance of $r_{\text{cut}}$ within a single MP step. Although multiple MP updates increase the effective cutoff distance, because information can "hop" between different neighborhoods as long as they share at least one atom, each MP step is accompanied by an undesirable loss of information, which limits the accuracy that can be obtained. Consequently, MPNNs are unable to describe non-local effects on length-scales that exceed the effective cutoff distance. To illustrate this problem, we consider the challenging open task [21] of learning the potential energy of cumulene molecules with different sizes (see Fig. 2a). Here, the relative orientation of the hydrogen rotors at the far ends of the molecule strongly influences its energy due to non-local electronic effects [21]. In order to be able to successfully learn the energy profile with a local model, the effective cutoff has to be large enough to allow information to propagate from one hydrogen rotor to the other.

As a representative example for MPNNs, we consider the recently proposed NEQUIP model [32], which achieves SOTA performance on several benchmarks. We find that even when the effective cutoff radius is large enough in principle, an MPNN with $n_l = 4$, $r_{\text{cut}} = 2.5\,\text{Å}$, and $l_{\max} \le 1$ fails to learn the correct energy profile. This is due to the fact that the relevant geometric information "cancels out" (similar to addition of vectors oriented in opposite directions) within each neighborhood,

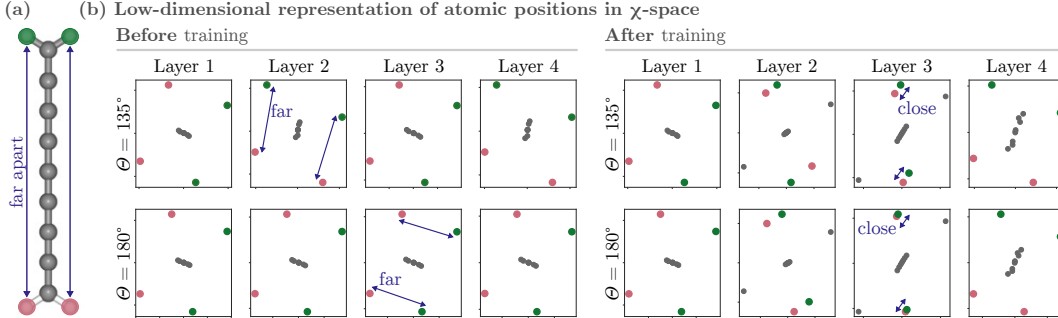

Figure 3: **(a)** Cumulene structure in Euclidean space. **(b)** Low-dimensional projections of the SPHCs ($l_{\max} = 1$) in SO3KRATES before and after training. Carbon atoms are grey and hydrogen atoms on different ends of the molecule are green and red, respectively. During training, the model maps hydrogen atoms at opposite ends close together in SPHC space (which is not the case at initialization).

underlining the limited expressiveness of mean-field interactions in MPNNs. Only by including higher-order geometric correlations, e.g. going to $l_{\max} = 3$, the correct energy profile can be recovered (at the cost of computational efficiency). When going to even larger cumulene structures, however, the effective cutoff becomes too small and it is necessary to increase the number of MP layers to solve the task (again, at the cost of lower computational efficiency), which is illustrated in appendix A.11. Neither increasing the maximum degree of interactions $l_{\max}$, nor the number of layers $n_l$, is a satisfactory workaround: Instead of offering a general solution to describe non-local interactions, both options decrease computational efficiency, while only shifting the problem to larger length-scales or higher-order geometric correlations.

We further apply three additional models to the cumulene structure with nine carbon atoms. To that end, we use an invariant SCHNET model with varying cutoff distances (6 Å and 12 Å), an inherently global but invariant sGDML model and the SPOOKYNET architecture which explicitly includes global effects using a non-local block. We find that none of the three is capable of describing the rotor energy profile of cumulene.

In contrast, our proposed SO3KRATES architecture is able to reproduce the energy profile for cumulene molecules of all sizes independent of the effective cutoff radius. Crucially, even with $l_{\max} = 1$, the predicted energy matches the *ab-initio* reference faithfully. We find that geometric corrections in the MP update of the SPHCs (cf. eq. (14)) are responsible for the increased capability of describing higher-order geometric correlations, as a SO3KRATES model with a naive MP update (cf. eq. (13)) fails to solve this task with $l_{\max} = 1$ (see Fig. 2a). We further confirm that the model picks up on the physically relevant interaction between the hydrogen rotors by analysing the attention values after training (see Fig. 8 appendix A.7). To illustrate how SO3KRATES is able to describe non-local effects, we show a low-dimensional projection of the atomic SPHCs before and after training for the largest of the cumulene molecules (Fig. 3). After training, the SPHCs for hydrogen atoms at opposite ends of the molecule are embedded close together in SPHC space, allowing SO3KRATES to efficiently model the non-local geometric dependence between the hydrogen rotors.

Generalization to structures, larger than those in the training data are usually associated with the re-usability of the learned, local representations. For that reason, it is unclear if this property still holds when non-local corrections are used. As we show in appendix A.5 a SO3KRATES model with non-local corrections still generalizes well to completely unknown and larger structures.

**Benchmarks, Data Efficiency and Generalization**  As pointed out in [31] and [32], equivariant features not only increase performance, but also improve data efficiency. The latter is particularly important, as *ab-initio* methods for reference data generation can become exceedingly expensive when high accuracy is required. Here, we use a subset of the recently introduced QM7-X data set [51], which we call QM7-X250. It contains 250 different molecular structures, each with 80 data points for training, 10 data points for validation and 11-3748 data points for testing (for details, see appendix A.9). The small number of training/validation samples per molecule makes it particularly suited for evaluating model behavior in the low data regime. In the following, we train (1) one model

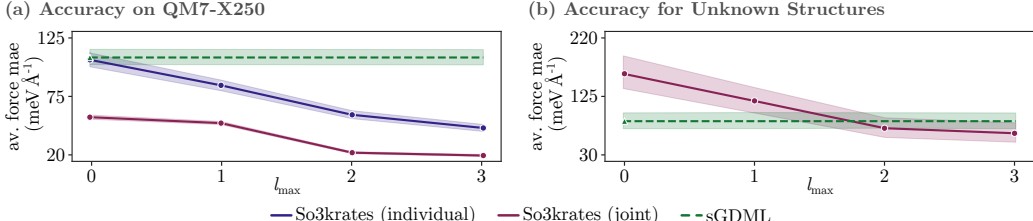

Figure 4: **(a)** Average force MAE as a function of $l_{\max}$ on 250 structures from QM7-X [51]. The blue solid line corresponds to SO3KRATES models trained on each of the 250 structures (à 80 training points), individually. The jointly trained SO3KRATES model (red solid line) is a single model trained on all $250 \times 80$ training points. **(b)** Accuracy of force predictions for unknown structures as function of $l_{\max}$. Generalization is only investigated for the jointly trained model. Due to the way the molecular descriptor is designed, SGDML [38] models can only be trained on individual structures.

per structure in QM7-X250 and (2) one model for all structures in QM7-X250 ($250 \times 80 = 20$k training points), which we refer to as *individual* and *joint* models, respectively.

We start by investigating the performance as a function of the maximal degree $l_{\max}$ and find that the error strongly decreases with higher $l_{\max}$ (Fig. 4a). As kernel methods are known to perform well in the low-data regime [21], we compare our results to SGDML [38] kernel models, which only use distances as a molecular descriptor (corresponding to $l_{\max} = 0$). For $l_{\max} = 0$, we find SGDML gives competitive results, whereas for $l_{\max} = 1$, SO3KRATES starts to outperform SGDML. As soon as $l_{\max} \geq 2$, however, the prediction accuracy of SO3KRATES is greater than that of SGDML by a large margin. Thus, increasing the order of geometric information in the SPHCs leads to strong improvements in the low-data regime. For jointly trained models, we find that SO3KRATES outperforms SGDML even for $l_{\max} = 0$, with continuous improvement for increasing $l_{\max}$. In appendix A.10 we report energy *and* force errors across degrees and further experimental details.

The generalization capability of SO3KRATES is tested, by applying a jointly trained model to 25 completely unknown molecules from the QM7-X data set (see, Fig. 4b, details in appendix A.10). Again, we find that force MAEs decrease with increasing $l_{\max}$. For reference, we compare SO3KRATES to individually trained SGDML models and find that SO3KRATES performs on par, or even slightly better, for $l_{\max} \geq 2$. Going beyond $l_{\max} = 2$ is found to only marginally improve generalization. In addition, we report results for a model trained on the full QM7-X data set in appendix A.5, following [30].

For completeness, we also apply SO3KRATES to the popular MD17 benchmark (see Table 1). We find, that SO3KRATES outperforms networks that have the same parameter complexity by a large margin (PAINN and NEWTONNET). Notably, it requires significantly less parameters than other SH based architectures (NEQUIP and SPOOKYNET), while performing only slightly worse or even on par with them. Furthermore, SO3KRATES outperforms DIMENET, its closest competitor in timing (cf. Fig. 1d), consistently by a large margin. Compared to current SH based approaches, GEMNETQ needs less parameters (still $\sim 2.5$x more than SO3KRATES) to achieve competitive results. However, it requires the explicit calculation of dihedral angles which scales cubically in the number of neighboring atoms. Due to its linear scaling (see A.8) and lightweight structure, SO3KRATES can significantly reduce the time for training and inference (see Fig. 1d and A.6).

## 5   Discussion and Conclusion

Due to the locality assumption used in most MPNNs, they are unable to model non-local electronic effects, which result in global geometric dependencies between different parts of a molecule. The length-scales of such interactions often greatly exceed the cutoff radius used in the MP step, and even though stacking multiple MP layers increases the effective cutoff, ultimately, MPNNs are not capable of efficiently modeling geometric dependencies on arbitrary length scales.

In this work, we contribute conceptually by proposing an efficient and scalable solution to this problem. We suggest a set of refinable, equivariant coordinates for point clouds in Euclidean space, called spherical harmonic coordinates (SPHCs). Non-local geometric effects can then be efficiently

Table 1: MAE for energy (in $\mathrm{kcal\,mol^{-1}}$) and forces (in $\mathrm{kcal\,mol^{-1}\,\mathring{A}^{-1}}$) of current state-of-the-art machine learning models on the MD17 benchmark for 1k training points. For each model, the number of paramaeters, as well as the scaling in the number of neighbors $m$ is shown.

| | | NEQUIP [32] 3M \| $\mathcal{O}(m)$ | SPOOKYNET [30] 3M \| $\mathcal{O}(m)$ | GEMNETQ [34] 2.2M \| $\mathcal{O}(m^3)$ | DIMENET [29] 1.9M \| $\mathcal{O}(m^2)$ | PAINN [31] 600k \| $\mathcal{O}(m)$ | NEWTONNET [52] 500k \| $\mathcal{O}(m)$ | SO3KRATES 700k \| $\mathcal{O}(m)$ |
|---|---|---|---|---|---|---|---|---|
| Aspirin | *energy* | 0.13 | 0.151 | – | 0.204 | 0.159 | 0.168 | 0.139 |
| | *forces* | 0.19 | 0.258 | 0.217 | 0.499 | 0.371 | 0.348 | 0.236 |
| Ethanol | *energy* | 0.05 | 0.052 | – | 0.064 | 0.063 | 0.061 | 0.052 |
| | *forces* | 0.09 | 0.094 | 0.088 | 0.230 | 0.230 | 0.211 | 0.096 |
| Malondialdehyde | *energy* | 0.08 | 0.079 | – | 0.104 | 0.091 | 0.096 | 0.077 |
| | *forces* | 0.13 | 0.167 | 0.159 | 0.383 | 0.319 | 0.323 | 0.147 |
| Naphthalene | *energy* | 0.11 | 0.116 | – | 0.122 | 0.117 | 0.118 | 0.115 |
| | *forces* | 0.04 | 0.089 | 0.051 | 0.215 | 0.083 | 0.084 | 0.074 |
| Salicyclic Acid | *energy* | 0.11 | 0.114 | – | 0.134 | 0.114 | 0.115 | 0.106 |
| | *forces* | 0.09 | 0.180 | 0.124 | 0.374 | 0.209 | 0.197 | 0.145 |
| Toluene | *energy* | 0.09 | 0.094 | – | 0.102 | 0.097 | 0.094 | 0.095 |
| | *forces* | 0.05 | 0.087 | 0.060 | 0.216 | 0.102 | 0.088 | 0.073 |
| Uracil | *energy* | 0.10 | 0.105 | – | 0.115 | 0.104 | 0.107 | 0.103 |
| | *forces* | 0.08 | 0.119 | 0.104 | 0.301 | 0.140 | 0.149 | 0.111 |

modeled by including geometric corrections, which are localized in the space of SPHCs, but non-local in Euclidean space. Further, we show that introducing spherical filter functions acting on the SPHCs increases geometric resolution and predictive accuracy.

We then propose the SO3KRATES architecture, a self-attention based MPNN, which decouples atomic features from higher-order geometric information. This allows to drastically decrease the parametric complexity while still achieving SOTA prediction accuracy. We show evidence that increasing the geometric order of SPHCs greatly improves model performance in the low-data regime, as well as generalization to unknown molecules.

A limitation of the current implementation of SO3KRATES is that spherical neighborhoods $\mathcal{N}_\chi$ in eq. (14) are computed from all pairwise distances in SPHC space. An alternative implementation could use a space partitioning scheme to find neighborhoods more efficiently. In a broader context, our work falls into the category of approaches that can help to reduce the vast computational complexity of molecular and material simulations. This can accelerate novel drug and material designs, which holds the promise of tackling societal challenges, such as climate change and sustainable energy supply [53]. Of course, our method could also be used for nefarious applications, e.g. design of chemical warfare, but this is true for all quantum chemistry methods.

Future research will focus on applications of SO3KRATES to materials and bio-molecules, which are typical examples of chemical systems where the accurate description of non-local effects is necessary to produce novel insights. Efficient treatment of non-local effects in point cloud data goes beyond the domain of quantum chemistry. One way of representing non-local dependencies are non-local neural networks [54]. In comparison to the presented approach they compute a relation in feature rather than in Euclidean space, making it incapable of capturing direct geometric relations in Euclidean space. However, this might be necessary if the relative orientation of objects far apart from each other plays a role for identifying different objects.

# 6   Acknowledgements

All authors acknowledge support by the Federal Ministry of Education and Research (BMBF) for BIFOLD (01IS18037A). KRM was partly supported by the Institute of Information & Communications Technology Planning & Evaluation (IITP) grants funded by the Korea government(MSIT) (No. 2019-0-00079, Artificial Intelligence Graduate School Program, Korea University and No. 2022-0-00984, Development of Artificial Intelligence Technology for Personalized Plug-and-Play Explanation and Verification of Explanation), and was partly supported by the German Ministry for Education and Research (BMBF) under Grants 01IS14013A-E, AIMM, 01GQ1115, 01GQ0850, 01IS18025A and 01IS18037A; the German Research Foundation (DFG). We thank Stefan Chmiela, Mihail Bogojeski and Nicklas Schmitz for helpful discussions and feedback on the manuscript.

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
