# A Appendix

## A.1 Proof of Equivariance

**SPHC Initialization** Here we give proof that certain quantities from the main text are invariant or equivariant, respectively. Let us start with the spherical harmonic coordinates (SPHC) which are initialized as

$$\boldsymbol{\chi}_i^{(l)}(R) = \frac{1}{\mathcal{C}_i} \sum_{j \in \mathcal{N}(i)} \phi_{r_{\text{cut}}}(r_{ij}) \cdot \boldsymbol{Y}^{(l)}(\hat{\boldsymbol{r}}_{ij}) , \tag{20}$$

where $\boldsymbol{Y}^{(l)} : S^2 \mapsto \mathbb{R}^{2l+1}$. In contrast to the main text, we make the dependence of the right hand side of eq. (20) on the atomic positions $\mathcal{R} = \{\boldsymbol{r}_1, \ldots, \boldsymbol{r}_n\}$ explicit. Each degree-wise entry in the initialized SPHCs (eq. (20)) transforms under rotation as

$$
\begin{aligned}
\boldsymbol{\chi}_i^{(l)}(R_\theta\, \mathcal{R}) &\sim \sum_{j \in \mathcal{N}(i)} \phi_{r_{\text{cut}}}(r_{ij}) \cdot \boldsymbol{Y}^{(l)}(R_\theta \hat{\boldsymbol{r}}_{ij}) \\
&= \sum_{j \in \mathcal{N}(i)} \phi_{r_{\text{cut}}}(r_{ij}) \cdot \boldsymbol{D}^l(R_\theta)\, \boldsymbol{Y}^{(l)}(\hat{\boldsymbol{r}}_{ij}) \\
&= \boldsymbol{D}^l(R_\theta) \sum_{j \in \mathcal{N}(i)} \phi_{r_{\text{cut}}}(r_{ij}) \cdot \boldsymbol{Y}^{(l)}(\hat{\boldsymbol{r}}_{ij}) \\
&= \boldsymbol{D}^l(R_\theta)\, \boldsymbol{\chi}_i^{(l)}(\mathcal{R}) ,
\end{aligned}
\tag{21}
$$

where $\boldsymbol{D}(R_\theta)^l \in \mathbb{R}^{(2l+1)\times(2l+1)}$ is the Wigner-D matrix for degree $l$ given rotation $R_\theta$ and $R_\theta \mathcal{R} = \{R_\theta \boldsymbol{r}_1, \ldots, R_\theta \boldsymbol{r}_n\}$. Since the cutoff function $\phi_{r_{\text{cut}}}$ takes the inter-atomic distance $r_{ij} = \|\boldsymbol{r}_j - \boldsymbol{r}_i\|_2$ as its input argument it is always invariant under rotation. Thus, each degree-wise entry $\boldsymbol{\chi}^{(l)}$ is equivariant after the initialization.

**SPHC Message Passing Update** After initialization, each per-degree entry in $\boldsymbol{\chi}$ is updated as

$$\boldsymbol{\chi}_i'^{(l)}(\mathcal{R}) = \boldsymbol{\chi}_i^{(l)}(\mathcal{R}) + \sum_{j \in \mathcal{N}(i)} \phi_{r_{\text{cut}}}(r_{ij}) \cdot \alpha_{ij}^{(l)} \cdot \boldsymbol{Y}^{(l)}(\hat{\boldsymbol{r}}_{ij}) , \tag{22}$$

where $\alpha_{ij}^{(l)}$ are rotationally invariant, per-degree self-attention coefficients. In the first layer, the first part in eq. (22) corresponds to the initialized SPHC entries, which have been already shown to be equivariant (see above). The second part of the equation has the same structural form as the initialization, with the additional self-attention value, which is a rotationally invariant scalar. Thus we can write

$$
\begin{aligned}
\boldsymbol{\chi}_i'^{(l)}(R_\theta\, \mathcal{R}) &= \boldsymbol{D}^l(R_\theta)\, \boldsymbol{\chi}^{(l)}(\mathcal{R}) + \sum_{j \in \mathcal{N}(i)} \phi_{r_{\text{cut}}}(r_{ij}) \cdot \alpha_{ij}^{(l)} \cdot \boldsymbol{Y}^{(l)}(R_\theta \hat{\boldsymbol{r}}_{ij}) \\
&= \boldsymbol{D}^l(R_\theta)\, \boldsymbol{\chi}^{(l)}(\mathcal{R}) + \boldsymbol{D}^l(R_\theta) \sum_{j \in \mathcal{N}(i)} \phi_{r_{\text{cut}}}(r_{ij}) \cdot \alpha_{ij}^{(l)} \cdot \boldsymbol{Y}^{(l)}(\hat{\boldsymbol{r}}_{ij}) \\
&= \boldsymbol{D}^l(R_\theta) \left( \boldsymbol{\chi}_i^{(l)}(\mathcal{R}) + \sum_{j \in \mathcal{N}(i)} \phi_{r_{\text{cut}}}(r_{ij}) \cdot \alpha_{ij}^{(l)} \cdot \boldsymbol{Y}^{(l)}(\hat{\boldsymbol{r}}_{ij}) \right) \\
&= \boldsymbol{D}^l(R_\theta)\, \boldsymbol{\chi}_i'^{(l)}(\mathcal{R}) ,
\end{aligned}
\tag{23}
$$

which shows that the updated $\boldsymbol{\chi}'^{(l)}$ is also equivariant. The proof of equivariance for later layers follows analogously.

**Transformation of the SPHCs** After having shown that each per-degree entry of $\boldsymbol{\chi}$ transforms under rotation according to the corresponding Wigner-D matrix $\mathbf{D}^l(R_\theta) \in \mathbb{R}^{(2l+1)\times(2l+1)}$, one can write the direct sum of all Wigner-D matrices as concatenation of matrices along the diagonal

$$\boldsymbol{D}(R_\theta) = \bigoplus_{l \in \mathcal{L}} \boldsymbol{D}^l(R_\theta) , \tag{24}$$

such that $\boldsymbol{D}(R_\theta) \in \mathbb{R}^{(l_{\max}-l_{\min}+1)\times(l_{\max}-l_{\min}+1)}$ has a block diagonal structure. The, full SPHC vectors transform under rotation as

$$\boldsymbol{D}(R_\theta)\,\boldsymbol{\chi}(\mathcal{R}) = [\boldsymbol{D}^{l_{\min}}(R_\theta)\boldsymbol{\chi}^{(l_{\min})}(\mathcal{R}),\dots,\boldsymbol{D}^{l_{\max}}(R_\theta)\boldsymbol{\chi}^{(l_{\max})}(\mathcal{R})]\,. \tag{25}$$

As each of the blocks along the diagonal of $\boldsymbol{D}(R_\theta)$ is an orthogonal Wigner-D matrix, $\boldsymbol{D}(R_\theta)$ itself is also orthogonal.

### A.1.1 Invariance of the (per-degree) Norm

The per degree norm is used as input to the spherical filter function $\phi_s$. As shown above, each of the per-degree entries in $\boldsymbol{\chi}$ transforms under rotation as

$$\boldsymbol{\chi}^{(l)}(R_\theta\,\mathcal{R}) = \boldsymbol{D}^l(R_\theta)\boldsymbol{\chi}^{(l)}(\mathcal{R})\,. \tag{26}$$

The squared norm can be expressed in terms of an inner product

$$
\begin{aligned}
\|\boldsymbol{\chi}^{(l)}(R_\theta\mathcal{R})\|_2^2 &= \left(\boldsymbol{\chi}^{(l)}(R_\theta\,\mathcal{R})\right)^T \boldsymbol{\chi}^{(l)}(R_\theta\,\mathcal{R}) \\
&= (\boldsymbol{D}^l(R_\theta)\,\boldsymbol{\chi}^{(l)}(\mathcal{R}))^T\,(\boldsymbol{D}^l(R_\theta)\,\boldsymbol{\chi}^{(l)}(\mathcal{R})) \\
&= \left(\boldsymbol{\chi}^{(l)}(\mathcal{R})\right)^T \underbrace{\left(\boldsymbol{D}^l(R_\theta)\right)^T \boldsymbol{D}^l(R_\theta)}_{=\mathbb{I}} \boldsymbol{\chi}^{(l)}(\mathcal{R}) \\
&= \|\boldsymbol{\chi}^{(l)}(\mathcal{R})\|_2^2,
\end{aligned}
\tag{27}
$$

where we use the orthogonality of Wigner-D matrices to show that the inner product is rotationally invariant. If $\|\cdot\|_2^2$ is invariant, so is $\|\cdot\|_2$, which completes the proof of equivariance for the degree-wise norm.

The squared norm of the full SPHC vector $\boldsymbol{\chi}$ transforms under rotation as

$$
\begin{aligned}
\|\boldsymbol{\chi}(R_\theta\,\mathcal{R})\|_2^2 &= \left(\boldsymbol{\chi}(R_\theta\,\mathcal{R})\right)^T \boldsymbol{\chi}(R_\theta\,\mathcal{R}) \\
&= \left(\boldsymbol{D}(R_\theta)\boldsymbol{\chi}(\mathcal{R})\right)^T \left(\boldsymbol{D}(R_\theta)\boldsymbol{\chi}(\mathcal{R})\right) \\
&= \left(\boldsymbol{\chi}(\mathcal{R})\right)^T \underbrace{\left(\boldsymbol{D}(R_\theta)\right)^T \boldsymbol{D}(R_\theta)}_{=\mathbb{I}} \boldsymbol{\chi}(\mathcal{R}) \\
&= \|\boldsymbol{\chi}(\mathcal{R})\|_2^2,
\end{aligned}
\tag{28}
$$

where we used that the orthogonality of $\boldsymbol{D}^l$ results in orthogonality of $\boldsymbol{D}$ (as argued above).

### A.2 Spherical Neighborhood

Starting point for the construction of the spherical neighborhood are the SPHCs $\mathcal{X}$ in a given layer of SO3KRATES. Consequently, the distance matrix in SPHC for all atomic pairs is given as an $n \times n$ matrix $\boldsymbol{X}$ with entries $\chi_{ij} = \|\boldsymbol{\chi}_i - \boldsymbol{\chi}_j\|_2$. The idea of a spherical neighborhood is to only consider atoms that lie within a certain distance w.r.t. each other in SPHC space, meaning only those for which $\chi_{ij} \leq \chi_{\text{cut}}$ holds. However, compared to the inter-atomic distances in Euclidean space, for which specific knowledge e.g. about bond lengths or interaction length scales exits, this is not the case for the rather abstract space of SPHCs. Thus, we apply a SOFTMAX function along each row (neighborhood) of $\boldsymbol{X}$, which gives a rescaled version of the spherical distance matrix $\tilde{\boldsymbol{X}}$ with rescaled entries $\tilde{\chi}_{ij} \in (0,1)$. Neighborhoods for each atom are then selected based on a polynomial cutoff function [29]

$$\phi_{\chi_{\text{cut}}}(x) = 1 - \frac{(p+1)(p+2)}{2}x^p + p(p+2)x^{p+1} - \frac{p(p+1)}{2}x^{p+2}, \tag{29}$$

where the input is given as $x = \tilde{\chi}_{ij}/\chi_{\text{cut}}$. Here we chose $\chi_{\text{cut}} = \kappa/n$, where $n$ is the number of atoms in the system and $\kappa$ allows to increase or decrease the size of the cutoff radius. By scaling with the inverse of the number of atoms in the system, we ensure that the relative number of neighboring atoms remains approximately constant, even when going to larger molecules. The parameter $\kappa$ allows to fine tune the relative number of atoms and can depended on the size and type of the molecule under investigation. For our cumulene experiments we chose $\kappa = 1$ and $p = 6$, which reduces the number of considered interactions by $\sim 40\%$ compared to a global model.

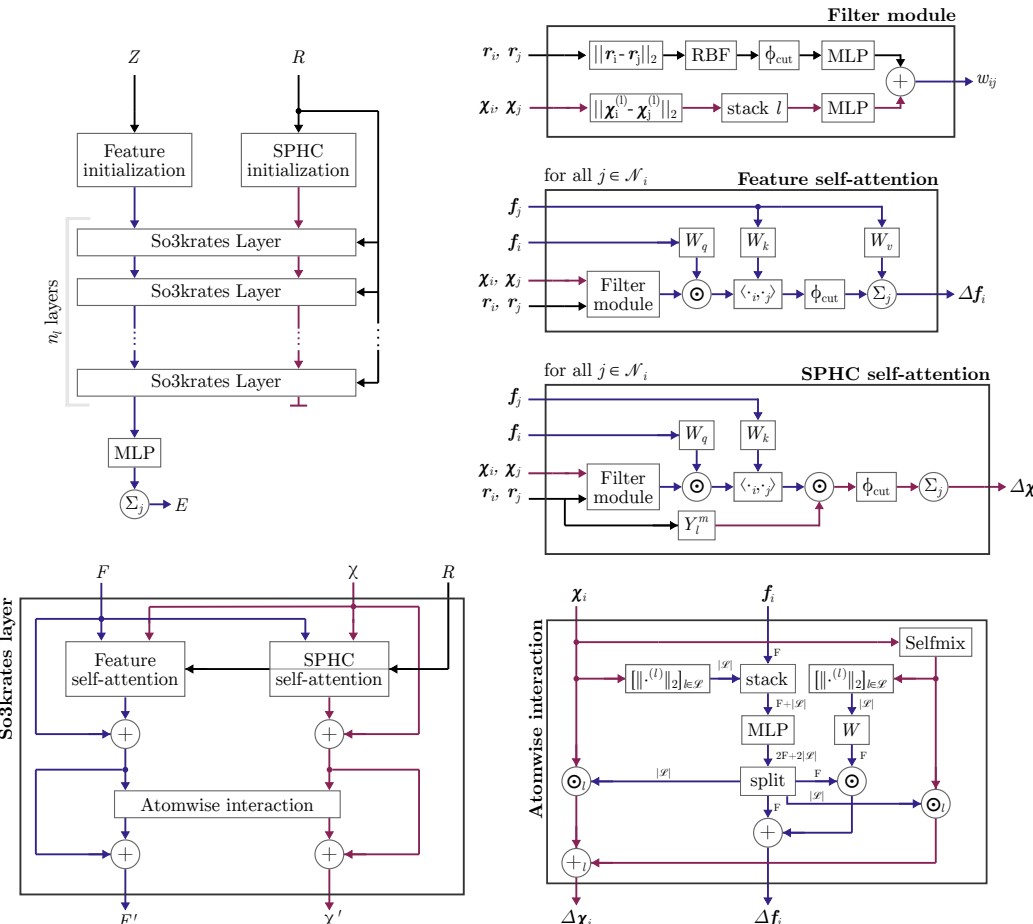

Figure 5: Figure shows the architecture of SO3KRATES and its individual building blocks. More details about the different modules can be found in the corresponding sections in A.3. Here $[\|\cdot l\|_2]_{l\in\mathcal{L}}$ denotes per-degree norm followed by concatenation. The subscript $l$ at the pointwise product and the addition (in the atomwise interaction module) illustrates that each degree is multiplied with one entry of the $|\mathcal{L}|$-sized output vector.

## A.3 Network Architecture

The SO3KRATES model is available at `https://github.com/thorben-frank/mlff`. It makes extensive use of the libraries FLAX [55] for building the networks, OPTAX [56] for training and NUMPY [57] and JAX [58] for additional processing steps. All code and data that is necessary to produce the results step by step presented in the main body of the paper can be downloaded from `https://zenodo.org/record/6584855` (newest version).

**Detailed Network Design** We now describe the specific implementation of the building blocks that we have formally introduced in the main text. The computational flow of the architecture is shown in Fig. 5.

**Radial Filter** For the radial filter function $\phi_r : \mathbb{R} \mapsto \mathbb{R}^F$ (first summand in eq. (16)) the interatomic distances are expanded in terms of $K$ radial basis functions as proposed in PHYSNET [19], which are given as

$$\theta^k(r_{ij}) = \phi_{r_{\text{cut}}}(r_{ij}) \cdot \exp\big(-\gamma(\exp(-r_{ij}) - \mu_k)^2\big), \tag{30}$$

where $\mu_k$ is the center of the $k$-th basis function ($k = 1 \ldots K$). The cutoff function

$$\phi_{r_{\text{cut}}}(r_{ij}) = \frac{1}{2}\left(\cos\left(\frac{\pi r_{ij}}{r_{\text{cut}}}\right) + 1\right) \tag{31}$$

guarantees that $\theta^k(r_{ij}) = 0$ smoothly goes to zero when interatomic distances exceed the cutoff radius $r_{\text{cut}}$. Here we chose $K = 32$ basis functions in total. The expanded distance is passed in a two-layered MLP with SILU non-linearity in-between and 128 hidden and $F$ output neurons, respectively.

**Spherical Filter**  The spherical filter function $\phi_s\left([\chi_{ij}^{(l)}]_{l\in\mathcal{L}}\right) : \mathbb{R}^{|\mathcal{L}|} \mapsto \mathbb{R}^F$ (second summand in eq. (16)) is also modelled as a two-layered MLP with the same non-linearity. Its input is given as the stacked per-degree distances $\chi_{ij}^{(l)}$, such that the input to the MLP is of low dimension (number of spherical degrees $l$). For that reason, we only use 32 neurons in the first and again $F$ neurons in the second layer.

**Self-Attention**  The self-attention matrix (cf. eq. (15)) is calculated from the filter vector $w_{ij}$ as well as from a pair of feature vectors $\boldsymbol{f}_i \in \mathbb{R}^F$ and $\boldsymbol{f}_j \in \mathbb{R}^F$. Before being passed to the inner product, each of the feature vectors is refined using a linear layer which we denote as $W_q$ and $W_k$ in the fashion of the key and query matrices usually appearing in the calculation of self-attention coefficients [49]. Thus, the self-attention coefficients are calculated as

$$\alpha_{ij} = \frac{1}{\sqrt{F}}\boldsymbol{q}_i^T(\boldsymbol{w}_{ij} \odot \boldsymbol{k}_j), \tag{32}$$

where $\boldsymbol{q}_i = W_q f_i \in \mathbb{R}^F$, $\boldsymbol{k}_j = W_k \boldsymbol{f}_j \in \mathbb{R}^F$ and $\boldsymbol{w}_{ij} \in \mathbb{R}^F$ comes from the filter module. The features $\boldsymbol{f}_j$ of the neighboring atoms are also passed through an additional linear layer $W_v$. The updated features are then given as

$$\boldsymbol{f}_i' = \boldsymbol{f}_i + \sum_{j\in\mathcal{N}(i)} \phi_{r_{\text{cut}}}(r_{ij}) \cdot \alpha_{ij} \cdot (W_v \boldsymbol{f}_j) \tag{33}$$

and the updates to the SPHCs as

$$\boldsymbol{\chi}_i'^{(l)} = \boldsymbol{\chi}_i^{(l)} + \sum_{j\in\mathcal{N}(i)} \phi_{r_{\text{cut}}}(r_{ij}) \cdot \alpha_{ij}^{(l)} \cdot \boldsymbol{Y}^{(l)}(\boldsymbol{r}_{ij}). \tag{34}$$

Different parameters are used for the feature and SPHC updates, respectively. For the feature update, we use a predefined number of heads whereas the number of heads in the SPHC update equals the number of degrees in the SPHC vector. In order to ensure permutation invariance, all parameters of the linear layers are shared across atoms.

**Atomwise Interaction**  After the update MP step, we update the features as well as the SPHCs per atom. In this step, we do not only include cross-degree coupling in $\boldsymbol{\chi}$ but also allow for information exchange between the feature and the SPHC branch. The functions $\phi_1\left(\boldsymbol{f}_i, [\chi_i^{(l)}]_{l\in\mathcal{L}}, [\tilde{\chi}_i^{(l)}]_{l\in\mathcal{L}}\right)$ and $\phi_2^{(l)}\left(\boldsymbol{f}_i, [\chi_i^{(l)}]_{l\in\mathcal{L}}, [\tilde{\chi}_i^{(l)}]_{l\in\mathcal{L}}\right)$ are implemented by a shared MLP. The function $\phi_3^{(l)}\left([\tilde{\chi}_i^{(l)}]_{l\in\mathcal{L}}\right)$ is implemented by a single linear layer, without bias term. The full computational flow is shown in Fig. 5.

## A.4   Ablation Study Spherical Filter

To illustrate the importance of the spherical filter, we examine its effect in an ablation study on the MD17 benchmark for varying maximal degree $l_{\text{max}}$. As can be seen in Fig. 6, using spherical filters (see eq. (16)) improves performance compared to a SO3KRATES model without them (solid vs. dotted lines), where the difference becomes even more pronounced for force predictions. Further, the effect of spherical filters becomes stronger for smaller $l_{\text{max}}$. Since many equivariant MPNNs carry features including additional channels for higher order geometric information, spherical filters can be straight forwardly integrated into current architectures, offering the potential of increased accuracy.

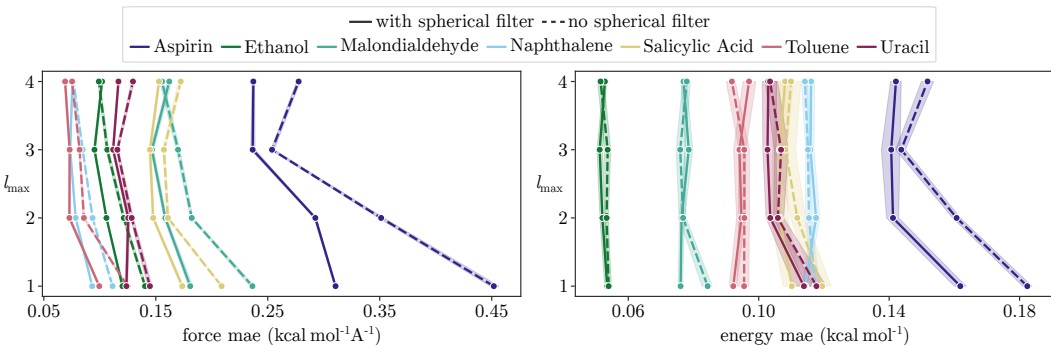

Figure 6: Ablation study on MD17 for using spherical filters. The model is evaluated on five different splits (a 1k points) of the data, where the transparent envelope is the $2\sigma$ confidence interval.

Table 2: Energy and force MAEs in meV and meV $\mathring{A}^{-1}$ for different models trained on the full QM7-X data set.

| | | SCHNET [13] | PAINN [31] | SPOOKYNET [30] | SO3KRATES $F = 132$ | SO3KRATES $F = 264$ | SO3KRATES + *non local* |
|---|---|---|---|---|---|---|---|
| known molecules / unknown conformations | *Energy* | 50.847 | 15.691 | 10.620 | 16.815 | 15.228 | 16.176 |
| | *Forces* | 53.695 | 20.301 | 14.851 | 20.422 | 18.446 | 19.879 |
| unknown molecules / unknown conformations | *Energy* | 51.275 | 17.594 | 13.151 | 21.733 | 21.750 | 20.071 |
| | *Forces* | 62.770 | 24.161 | 17.326 | 25.211 | 23.169 | 24.236 |

## A.5 QM7-X Experiments

As an additional benchmark, we train SO3KRATES on the full QM7-X data set and compare our results to the reported errors for known and unknown molecules in table 2. The known molecules correspond to the test samples that have not been seen during training. Following [30], we use the keys (IDMOL in the QM7-X data base) [1771, 1805, 1824, 2020, 2085, 2117, 3019, 3108, 3190, 3217, 3257, 3329, 3531, 4010, 4181, 4319, 4713, 5174, 5370, 5580, 5891, 6315, 6583, 6809, 7020] for testing the generalization of our model to unknown structures, which have been excluded from the data set prior to generating train, test and validation splits.

**Accuracy Comparison Local vs. Non-Local Model** By comparing the results reported in table 2, we find, that the non-local model shows the same accuracy as the local model for both known and unknown molecules. This underlines the applicability of the presented, non-local corrections to a large variety of molecular structures including transfer learning to unknown molecules.

**Accuracy Improvement by Up-Scaling** Despite the parametric leightweight structure of SO3KRATES, it is important that results can be imrproved by upscaling the model. Here we take one of the most straight forward paths and upscale the model by simply increasing the feature dimension from $F = 132$ to $F = 164$. As it can be seen in table 2, this already allows to increase the accuracy on the QM7-X dataset compared to the base line model. It should be noted, however, that one could follow additional/other directions, such as increasing the maximal degree $l_{\max}$.

**Generalization to Larger Molecules with a Non-Local Model** As seen in table 2, a model with non-local correction is capable of generalizing to structures not seen during training. However, the re-usability of local information is one of the key assumptions for building models that can be trained on small molecules and afterwards be applied to larger, completely unknown structures [30]. Thus, testing the non-local model on unknwon structures from the QM7-X data set is an insufficient test to investigate this property of transfer learning. We therefore use the non-local model for geometry optimization of molecules that range from 47 (riboflavin) up to 109 atoms (Ala10) in size (see 7). We find that the non-local model is robust for geometry optimization when being applied to much larger, unknown molecules. The largest molecule in the QM7-X data set has 23 atoms.

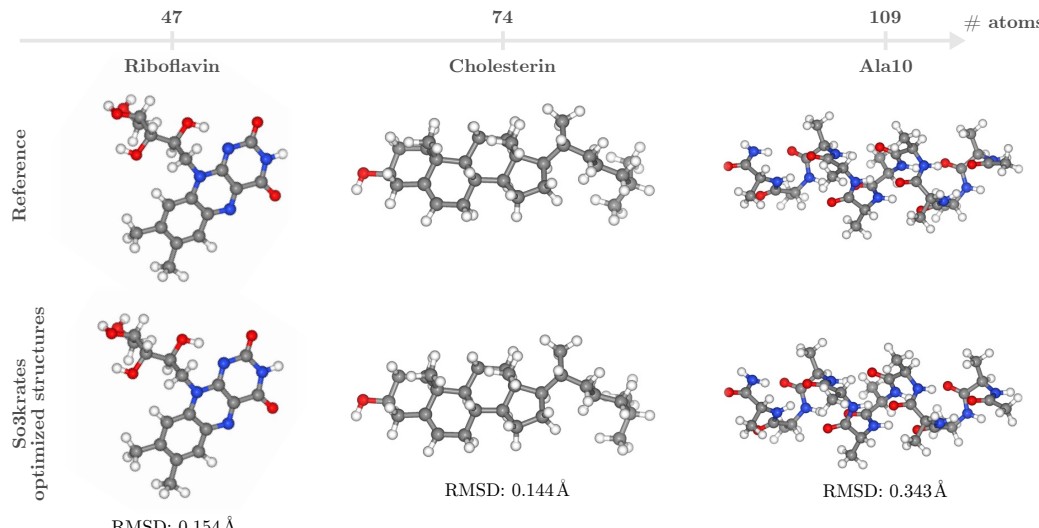

Figure 7: The upper row shows the optimized reference geometry obtained with PBE0-MBD calculation (taken from [30]). The bottom row shows geometry optimized structures obtained with a SO3KRATES model *with* non-local corrections from SPHC space as well as the root mean squared deviation from the reference structure. The model has been trained on the QM7-X data set for which the largest contained structure has 23 atoms. Thus, SO3KRATES can generalize to larger, unknown structures without relying on the assumption of locality.

Table 3: Training and inference times for different models for the toluene molecule from the MD17 benchmark and a batch size of 4. Hyperparemters for SO3KRATES are the ones that have been used to produce the reported results on the MD17 benchmark (cf. Tab. 1). Times for DIMENET and GEMNETQ have been measured on a different GPU, such that we decrease their runtimes by the factor reported here [37]. Inference times are for energy and force predictions.

|  | SO3KRATES | | NEQUIP [32] | | DIMENET [29] | | GEMNETQ [34] | |
|  | *training* | *inference* | *training* | *inference* | *training* | *inference* | *training* | *inference* |
| time (ms) | 34 | 12 | 507 | 136 | 218 | 24 | 483 | 76 |

## A.6 Time Analysis

In order to determine the training and inference times, we follow [34] and evaluate our model on the toluene molecule from the MD17 benchmark with a batch size of 4. We trained a NEQUIP model with the same hyperparameters as reported in [32] as well as a SO3KRATES model on a Tesla P100 with 12GB. The reported training time corresponds to the wall time it took each model to evaluate a single gradient update (without time for validation). We compare these runtimes to the runtimes that have been reported for DIMENET [29] and GEMNETQ [34] in [34]. However, these reported times have been measured on a GeForce GTX 1080Ti (a GPU we did not have access to). In [37] it has been found that a Tesla P100 gives a speedup factor of $\sim 1.3$, such that we downscale the reported runtimes accordingly. The resulting times are shown in table 3, which are the values plotted in 1d. It should be noted, that our implementation did not focus on the runtime, such that it is likely to be possible to further reduce the computational cost that is required for training and inference.

## A.7 Analysis of Attention Coefficients

In figure 8 we plot the attention coefficients from the MP update of the SPHCs (cf. eq. (14)) after training for different dihedral angles between the rotors. Attention values are calculated as the average over all attention values obtained for a given atomic pair throughout all SPHC update steps. As it can be verified visually, the model picks up physically important interactions. Care should be taken,

**Attention Coefficients**

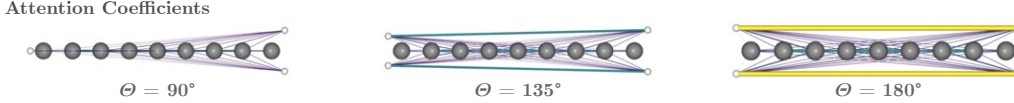

$\Theta = 90°$       $\Theta = 135°$       $\Theta = 180°$

Figure 8: Visualization of the attention coefficients from the MP update of the SPHCs (cf. eq. (14)) for different dihedral angles between the rotors.

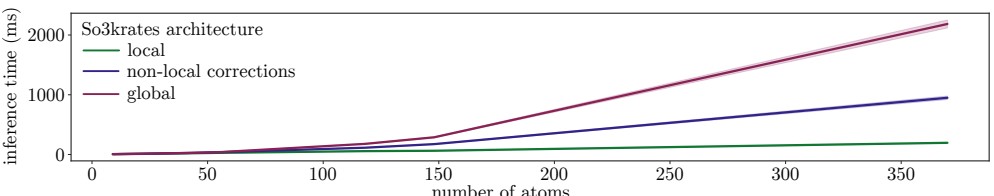

Figure 9: Scaling comparison between a local SO3KRATES (as used in the first part of the experiment section 4.1), an architecture with non-local corrections (as used in the second part of the experiment section 4.2) and a global SO3KRATES model (only used for the sake of comparison). As expected, the local model shows linear scaling in the number of atoms and only takes $\sim 196\,\mathrm{ms}$ for a system with 370 atoms. We also find the model with non-local correction to balance between the quadratically scaling global model and the linear local model. Due to memory issues for the fully global model for larger structures, we only report inference times for energy predictions.

however, since it has been pointed out in [59] that looking at the bare attention values only has limited expressiveness.

## A.8 Scaling Analysis

We further compare the scaling of three different versions of the SO3KRATES model. Namely, we compare a local SO3KRATES model, which only localizes in $\mathbb{R}^3$ (used in 4.1), a non-local SO3KRATES model with non-local corrections from SPHC space (used in 4.2) and a fully global model for which the cutoff radius is chosen such that all atoms are in each others neighborhood. The inference times for energy predictions for molecules ranging from 9 up to 370 atoms are shown in Fig. 9. As expected, we find linear scaling for the local SO3KRATES version with a remarkable inference time of only $\sim 196\,\mathrm{ms}$ for 370 atoms (batch size is 25). For the fully global mode, we see quadratic scaling in the number of atoms. The model with non-local corrections can be found somewhere in-between the local and the global model. To that end, we want to stretch the fact that we did not focus on an efficient implementation for the non-local corrections. The experiments were run on a Tesla P100 with 12 Gb and a batch size of 25.

## A.9 QM7-X250

As starting point for the recently introduced QM7-X dataset [51] serve $\sim 7k$ molecular graphs with a maximum of 7 non-hydrogen atoms (C, N, O, S, Cl). By sampling and optimizing structural and constitutional isomers for each graph, $\sim 42k$ equilibrium structures are generated. Using normal mode sampling at 1500 K, 100 out-of-equilibrium points are generated for each structure resulting in 101 data points per structure and $\sim 4.2M$ geometries in total.

In order to make the data set well suited for both, kernel and neural network models, we group the geometries by structural isomers which gives $\sim 13k$ individual data sets, each consisting of *#stereo-isomers* $\times 101$ geometries. For each of the data sets we choose 80 points for training, 10 points for validation and the remaining points for testing after the training. Afterwards, we randomly sample 250 data sets out of the 13k data sets. The comparison of the probabilities of drawing a molecule with a given number of atoms, number of symmetries and number of stereo-isomers from the original QM7-X and the QM7-X250 data sets are shown in Fig. 10. Since we ensure that each of the structural subsets present in the original data set is also present in the 250 drawn samples at least once, it can be seen that these structures are over represented in the QM7-X250 data set, even though

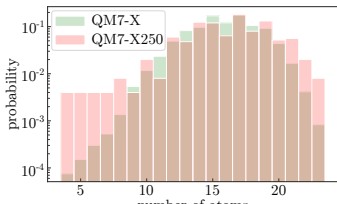 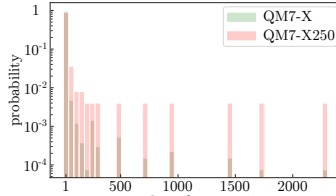 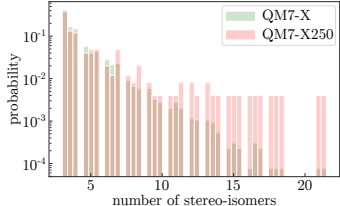

Figure 10: The figure shows the probability of occurrence for structures with a certain property value in the original QM7-X and the sub-sampled QM7-X250 dataset. As it can be seen, QM7-X follows the distribution of the original dataset. As for each property value at least once structure is present in QM7-X250 (per design), values with a low count in the original data set have higher relative importance, which can be seen e.g. for structures with only a few atoms, or with high symmetry.

Table 4: Comparison of the averaged per structure MAE for energy (in meV) and forces (in meV $\text{Å}^{-1}$) for sGDML and SO3KRATES models with varying $l_{\max}$. Best results are in bold. Note that for small $l_{\max}$ the total number of parameters decreases, since in the SPHC update step, as many heads are used as there are degrees (one degree for $l_{\max} = 0$ and $l_{\max} = 1$). As a consequence, the size of the matrices $Q$ and $K$, that are applied to the feature vectors that make up the inner product, become smaller in size. The feature dimension $F = 132$ is the same for all models. This leads to a decrease of the total number of network parameters, which is larger than the increase in network parameters, due to the additional degrees. Note that it is important to take the average over per structure MAEs, since the different structures have differently many test samples (see section A.9).

| | | sGDML [38] | | SO3KRATES $l_{\max} = 0$ | | SO3KRATES $l_{\max} = 1$ | | SO3KRATES $l_{\max} = 2$ | | SO3KRATES $l_{\max} = 3$ | |
| | | *individual* | *joint* | *individual* | *joint* | *individual* | *joint* | *individual* | *joint* | *individual* | *joint* |
|---|---|---|---|---|---|---|---|---|---|---|---|
| QM7-X250 | *energy* | 67.78 | – | 78.40 | 46.87 | 66.43 | 44.64 | 47.02 | 19.10 | **38.40** | **17.09** |
| | *forces* | 107.66 | – | 105.80 | 57.57 | 84.54 | 52.58 | 59.33 | 27.77 | **48.46** | **25.37** |
| # parameters | | – | | 846k | | 846k | | 746k | | 716k | |

they only make up a single structure. Apart from these special structures, we see that the sampled dataset correctly reproduces the distribution of the original data set.

From the 250 sampled structures, we build 250 per structure data sets with 80 training, 10 validation and 10–3748 (depending on the number of stereo-isomers) test points, which we referred to as *individual* dataset in the main text. We can further build a *joint* dataset, by merging all individual structures into a single data set, which gives $20\,000$ ($250 \times 80$) training, $2\,500$ ($250 \times 10$) validation, and $108\,800$ testing points.

## A.10   QM7-X250 Experiments

In the main text we show the force MAE as a function of maximal order $l_{\max}$ in Fig. 4a. In table 4, we further show the exact errors for energy and forces as a function of maximal order $l_{\max}$, as well as the number of parameters per SO3KRATES model. For all models with $l_{\max} \geq 1$, we do not include $l_{\max} = 0$ within the SPHCs, since the zeroth degree evaluates to a constant one and thus does not contain any additional geometric information. The generalization to unknown molecules is tested on the same structure keys as in the SPOOKYNET paper [30] and in the QM7-X experiment from above A.5. As we only trained on forces and generalize to completely unknown molecules, we can not fit the energy integration constant (as we assume to have no reference data for the unknown molecules). For that reason we only report force errors (see 5). However, one could train a SO3KRATES model on both, energy and forces to obtain meaningful predictions for both. In that case, atomization energies would need to be included to obtain equal energy scales across different molecules as done for the full QM7-X data set.

Table 5: Comparison of the averaged per structure MAE for forces (in meV $\text{Å}^{-1}$) for sGDML and SO3KRATES models with varying $l_{\max}$ when applied to completely unknown structures. Best results are in bold. Due to the way the molecular descriptor is designed, sGDML [38] models can only be trained on individual structures. Thus the reported results for sGDML are not generalization results but rather serve as a benchmark for the generalization MAE of SO3KRATES. The number of parameters are the same as reported in Tab. 4. Note that it is important to take the average over per structure MAEs, since the different structures have differently many test samples (see section A.9).

| | | sGDML [38] | SO3KRATES $l_{\max} = 0$ | SO3KRATES $l_{\max} = 1$ | SO3KRATES $l_{\max} = 2$ | SO3KRATES $l_{\max} = 3$ |
|---|---|---|---|---|---|---|
| Generalization | *forces* | 86.84 | 159.42 | 117.91 | 76.05 | **68.29** |

Description of Non-Local, Geometric Interactions in Cumulene for different number of MP steps

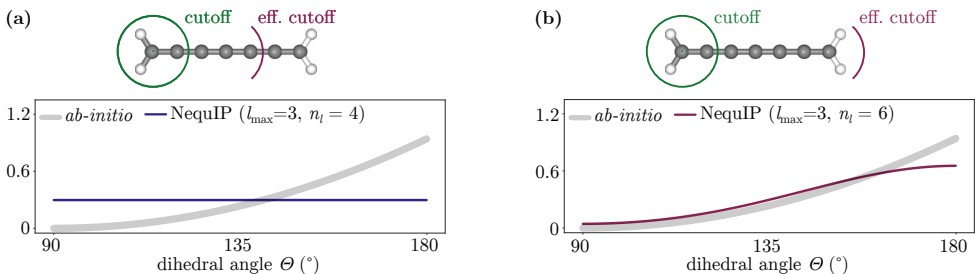

Figure 11: **(a)** Description of the energy profile in cumulene using $n_l = 4$ in the NEQUIP model, such that the effective cutoff radius is smaller then the length scale on which the electronic effects take place. **(b)** By increasing the number of layers to $n_l = 6$, the effective cutoff becomes large enough to transfer information about the rotor orientation along the molecular graph. However, the problem is only shifted to larger length scales.

### A.11 Additional Experiments for Non-Local Effects

Here we demonstrate, that increasing the number of local MP steps in NEQUIP allows to model the non-local effects in cumulene, as an increase of the number of steps increases the effective cutoff of the model (see Fig. 11). However, this just shifts the problem to larger distances; increasing the length of the cumulene molecule again leads to a scenario where the local model fails.

### A.12 Benchmarks for Non-Local Effects

We further apply SO3KRATES to the benchmark presented in [60], which explicitly introduces structures which exhibit non-local effects. We find, that for the carbon chain with non-local charge transfer, SO3KRATES with non-local corrections improves by a factor $\sim 2$ compared to a local model. For the remaining structures no such difference can be found. Notably, the non-local effects from the presented benchmark do not result in global geometric relations but rather change the overall behavior by adding or removing certain atoms (called doping). Thus, a combination of the SO3KRATES mechanism (for geometric relations) and the mechanism presented in SPOOKYNET might be a valuable direction for future work. Full results are reported in table 6. Note that SO3KRATES could be easily extended to predict partial charges as well following e.g. [30].

### A.13 Training Details

We train SO3KRATES by minimizing a combined loss of energy and forces

$$\text{Loss} = (1 - \beta) \cdot (E - \tilde{E})^2 + \frac{\beta}{3N} \sum_{k=1}^{n} \sum_{i \in (x,y,z)} (F_k^i - \tilde{F}_k^i)^2, \tag{35}$$

Table 6: RSME errors for energy and forces in meV / atom and meV / Å for various generations of Behler-Parinello networks [60], SPOOKYNET [30] and a local and non-local SO3KRATES model.

| | | 2G-BPNN | 3G-BPNN | 4G-BPNN | SPOOKYNET | SO3KRATES | SO3KRATES + *non local* |
|---|---|---|---|---|---|---|---|
| $C_{10}H_2/C_{10}H_2^+$ | *Energy* | 1.619 | 2.045 | 1.194 | 0.364 | 0.113 | 0.122 |
| | *Forces* | 129.5 | 231.0 | 78.0 | 5.802 | 13.198 | 7.844 |
| | *Charges* | – | 20.08 | 6.577 | 0.117 | – | – |
| $Na_{8/9}Cl_8^+$ | *Energy* | 1.692 | 2.042 | 32.78 | 1.052 | 0.455 | 0.474 |
| | *Forces* | 57.39 | 76.67 | 15.82 | 1.052 | 3.126 | 3.316 |
| | *Charges* | – | 20.80 | 1.323 | 0.111 | – | – |
| $Au_2$-MgO | *Energy* | 2.287 | – | 0.219 | 0.107 | 0.062 | 0.064 |
| | *Forces* | 153.1 | – | 66.0 | 5.337 | 8.130 | 9.472 |
| | *Charges* | – | – | 5.698 | 1.013 | – | – |

Table 7: **Training hyperparameters:** Table summarizes the training hyperparameters, used for the different models and experiments. Experiments are identified via their figure/table number in the main text. For the results reported in Tab. 6, the same sizes of data splits as reported in [60] have been used (indicated by an asterix).

| Ref. | $F$ | $n_{\text{layers}}$ | $r_{\text{cut}}$ (Å) | $l_{\max}$ | geom. corr. | spherical filter | $\beta$ | epochs | $B_s$ | $N_{\text{train}}$ | $N_{\text{valid}}$ |
|---|---|---|---|---|---|---|---|---|---|---|---|
| Fig. 2 | 128 | 4 | 2.5 | 1 | True/False | True | 1 | 4k | 8 | 1k | 1k |
| Fig. 3 | 128 | 4 | 2.5 | 1 | True/False | True | 1 | 4k | 8 | 1k | 1k |
| Fig. 4a | 132 | 6 | 5 | [0,1,2,3] | False | True | 1 | 1.5k | 1 | 80 | 10 |
| Fig. 4b | 132 | 6 | 5 | [0,1,2,3] | False | True | 1 | 6k | 100 | 20k | 2.5k |
| Tab. 1 | 132 | 6 | 5 | 3 | False | True | 0.99 | 4k | 8 | 1k | 1k |
| Fig. 6 | [128, 128, 132, 128] | 6 | 5 | [1,2,3,4] | False | True / False | 0.99 | 4k | 8 | 1k | 1k |
| Fig. 7 | 132 | 6 | 5 | 3 | True | True | 0.99 | 1k | 100 | 3.6M | 360k |
| Tab. 2 | 132 | 6 | 5 | 3 | True/False | True | 0.99 | 1k | 100 | 3.6M | 360k |
| Tab. 2 | 264 | 6 | 5 | 3 | False | True | 0.99 | 1k | 100 | 3.6M | 360k |
| Tab. 6 | 132 | 6 | 5 | 3 | True/False | True | 0.99 | 1k | 100 | * | * |

where $\tilde{E}$ and $\tilde{F}$ are the ground truth and $E$ and $F$ are the predictions of the model. The loss is evaluated on mini batches with a batch size given in table Tab. 7. The parameter $\beta$ is used to control the trade-off between energy and forces and additionally accounts for different energy and force scales. We train our models with the ADAM optimizer [61] and an initial learning rate of $\mu = 1 \times 10^{-3}$. We use exponential learning rate decay where the learning rate is decreased by a factor of $0.5$ every 1k epochs for the MD17 benchmark and the joint QM7-X250 dataset and every 300 epochs for individual models on the QM7-X250 dataset. For training on the full QM7-X data set we reduced the leraning rate every 250 epochs by a factor of 0.7. We further applied gradient clipping to a maximal norm of 1.

Additional hyperparameters that have been used to produce the tables and figures in this work are given Tab. 7. Whenever $\beta = 1$ is reported, no energy contribution did enter the loss function. In that case, we calculated the integration constant for energy according to section A.14.

## A.14 Energy Integration Constant

When only training on forces, the resulting energy predictions are likely to be shifted w. r. t. the correct energy values, due to vanishing constants when taking the derivative. Since force fields are conservative vector fields, one can define the following loss for the constant $c$ as

$$
\begin{aligned}
\mathcal{L}(c) &= \sum_{i=1}^{M} \left( \int \mathbf{f}_{\text{F}}(R_i) \, \mathrm{d}R_i - E_i \right)^2 \\
&= \sum_{i=1}^{N_{\text{data}}} \left( f_{\text{E}}(R_i) + c - E_i \right)^2
\end{aligned}
\tag{36}
$$

where index $i$ runs over the $M$ data points, $R_i \in \mathbb{R}^{n \times 3}$ are the atomic coordinates and $E_i$ is the reference value of the PES. The functions $\mathbf{f}_{\mathrm{F}} : \mathbb{R}^{n \times 3} \mapsto \mathbb{R}^{n \times 3}$ and $f_{\mathrm{E}} : \mathbb{R}^{n \times 3} \mapsto \mathbb{R}$ are the force and energy function, respectively. Minimization w. r. t. to $c$ then gives

$$\partial_c \mathcal{L}(c) \stackrel{!}{=} 0 \tag{37}$$

$$\Longleftrightarrow \qquad 2 \sum_{i=1}^{M} c - \big( E_i + f_{\mathrm{E}}(R_i) \big) = 0 \tag{38}$$

$$\Longleftrightarrow \qquad \frac{1}{M} \sum_{i=1}^{M} E_i + f_{\mathrm{E}}(R_i) = c \,. \tag{39}$$

Thus, the shifted energy function is given as $\tilde{f}_{\mathrm{E}}(R_i) = c + f_{\mathrm{E}}(R_i)$.