# OpenReview forum: "So3krates: Equivariant attention for interactions on arbitrary length-scales in molecular systems"
_NeurIPS.cc/2022/Conference — NeurIPS 2022 Accept_

### Official Review · Reviewer_jF7P · 2022-07-11

**Rating:** 7
**Confidence:** 4
**Soundness:** 3 good
**Presentation:** 3 good
**Contribution:** 3 good

**Summary:**

The paper proposes to use separate representations of spherical harmonic coordinates
jointly with representations of atomic features in learning models of molecular force fields.
This allows message passing to local neighborhoods in the spherical harmonic coordinates
which translate to non-local neighborhoods in Euclidean space. As a result the method
is able to model long-range interaction effects that are prohibitively expensive to model with
existing, hop-limited message-passing graph neural networks.  It is also helpful to the
low-data cases since the fewer number of parameters in the network requires less training
data to estimate.

**Questions:**

The idea of using harmonic functions like wavelets in deep learning has been
pursued before [for example, see "Wavelet Neural Networks:
With Applications in Financial Engineering, Chaos, and Classification",
Antonios K. Alexandridis, Achilleas D. Zapranis, John Wiley & Sons, 2014].
How does the current proposal compare to those uses?


**Limitations:**

The authors discuss the limits of the current applications to materials
discovery, while projecting a wider scope of application to other spatial modeling needs
like in computer vision.  However, in those wider areas, there have been more active
considerations of other representations to capture spatio-temporal dependences.
Some review of where the proposal approach could distinguish itself can add to the
argument.


**Strengths And Weaknesses:**

The proposal is sound and convincing, and echoes the long established methods
in signal processing.  Experimental demonstration and comparison to prior art are convincing.

---

> ### Author Response · Authors · 2022-08-02
> **Authors response to the reviewer**
>
> We thank the reviewer for their constructive criticism.
>
> In particular, we are grateful for making us aware of wavelet neural networks and their applications in financial engineering, chaos, and classification. Although these domains seem to differ from quantum chemistry systems at first glance, all fields require flexible and expressive models, which benefit from the usage of periodic functions. We will add a discussion of these similarities and a citation to the paper by Alexandridis et al. (mentioned by the reviewer) to the camera-ready version.
>
> Further, we will include a discussion of other modeling approaches for (long-range) spatio-temporal dependencies in the field of computer vision. As a representative example, we consider non-local neural networks [https://ieeexplore.ieee.org/document/8578911] which are one way of representing non-local dependencies. In comparison to our approach, non-local neural networks rely on relations between feature representations. Loosely speaking, they compute a relation in feature space rather than in Euclidean space. Thus, a non-local neural network is not capable of capturing direct geometric relations in Euclidean space. This might be necessary if the relative orientation of objects far apart from each other plays a role for identifying different objects. We will add a detailed corresponding discussion to the final version of the manuscript.
>
> We hope that these proposed changes adequately address the concerns raised by the reviewer.

---

### Official Review · Reviewer_SNvt · 2022-07-11

**Rating:** 6
**Confidence:** 3
**Soundness:** 2 fair
**Presentation:** 3 good
**Contribution:** 2 fair

**Summary:**

The authors propose a new ML force field based on higher order interactions and equivariant self-attention. The model represents atomic features using spherical harmonic coordinates that are updated during the forward pass through the network, and spherical neighborhoods are constructed using these representations. This helps the model learn relations between distant atoms. As a result, the proposed method is computationally efficient, requires few parameters and achieves comparable performance to recent methods like Nequip.

**Questions:**

1. How does the performance vary with the number of nodes? Spherical neighborhoods computation and self-attention could be very expensive for large systems like bio-molecules and materials.
2. MD17 is a small, relatively homogenous dataset. Do the authors have a sense for how the model would perform on a large and diverse dataset like OC20 (opencatalystproject.org)?


**Limitations:**

Seems adequate.

**Strengths And Weaknesses:**

Strengths
* SPHC representation helps the model capture global dependencies in a natural way.
* The entire model is equivariant, which is a desirable property to have for its data efficiency.
* The proposed method requires very few parameters to achieve good performance.
* It is also very fast for inference, which is an important property to have for many applications like MD simulations where millions of inference steps are needed.

Weaknesses
* The paper does not present results from multiple, diverse datasets (unlike many other papers in this area). So, it is unclear how generalizable the results are.
* The results in table 1 show that Nequip obtains better results (though at higher training & inference cost). Therefore, this method is only suitable when computational efficiency is more important than accuracy.

---

> ### Author Response · Authors · 2022-08-02
> **Authors response to the reviewer**
>
> We thank the reviewer for their positive assessment and helpful comments.
>
> As a weakness of our paper, the reviewer mentions the lack of diversity of the presented results. To address this point, we performed three additional experiments, including a benchmark on the full QM7-X dataset (as done in the SpookyNet paper) and experiments on data sets introduced in a paper by Ko et al. (suggested by reviewer toeX), which includes small molecules with varying charge states, and the interaction between a doped and undoped material with gold atoms. Further, we show that our model generalizes to larger and unknown structures such as a polypeptide consisting of ten amino acids (see answer to reviewer toeX). We politely disagree with the notion that So3krates is only suitable if computational efficiency is more important than accuracy. Rather, there is a tradeoff between training/inference time and achievable accuracy. However, we admit that we did not show or discuss this point in our original submission, so we fully understand why the reviewer got this impression. We put the primary focus on low inference times (rather than highest possible accuracy) in our original submission, but we are currently performing additional experiments for So3krates models with increased complexity. The results of these additional runs, in particular with respect to which accuracy can be achieved by simply scaling up the architecture, will be included in the camera-ready version.
>
>
> The reviewer also raised two questions regarding (a) scalability and (b) performance on large and diverse data sets:
>
> We performed an additional experiment to investigate the scaling behavior with respect to the number of nodes/atoms. Preliminary results can be found in our reply to reviewer toeX and the final results will be included in the camera-ready version.
>
> We agree that the performance of So3krates on less homogenous and more challenging datasets than MD17 is an important point, which is why we performed several additional experiments (see above). Given the performance of So3krates on these additional datasets, we expect to also observe good performance on the OC20 dataset. Unfortunately, due to the size of the OC20 dataset, we cannot perform this benchmark with our current setup. We are currently applying for additional computational resources and hope that we can include results for OC20 in the camera-ready version of the manuscript.
>
> Should the proposed changes adequately address the concerns raised by the reviewer, we kindly ask them to consider raising their score accordingly.

---

### Official Review · Reviewer_toeX · 2022-07-19

**Rating:** 7
**Confidence:** 4
**Soundness:** 3 good
**Presentation:** 3 good
**Contribution:** 3 good

**Summary:**

The authors consider the problem of capturing non-local effects in molecular modelling. To address this, they construct a model based on:

- Spherical Harmonic Coordinates (SPHC): A wavelet transform with different orders of spherical harmonic is used to represent the molecule geometry in a SPHC space. An equivariance relation between rotations in the euclidean input and Wigner-D transformations in SPHC gives the model the right symmetries, and the intention is that non-local interactions in Euclidean space may become local in SPHC.
- An adapted message passing scheme which passes messages in local neighbourhoods both in Euclidean and SPHC space while preserving equivariance.

The authors provide demonstration of their method on an educational example of cumulene twisiting as well as larger benchmarking on QM7-X250 and MD17.

**Questions:**

The proposed SPHC solution is well suited to the specific angle-dependent challenge of cumulene, however there are no other examples of non-local interaction such as dipole-dipole or charge-dipole interaction highlighted in the paper. Is there another case study where SPHC is the correct representation to capture the non-local effects and a 6A cutoff is so clearly deficient? How does So3krates do at modeling water clusters or the Au2-MgO system considered in the SpookyNet paper or other examples originating from Ko et al?

**Limitations:**

Locality is often built-in to neural-network based force fields to both (1) limit computational cost and (2) enable generalisation from smaller sub-systems to larger simulations. Do we lose these when we use a non-local model? To what extent can So3krates be scaled to something like the 100 million atom system recently run on 128 GPUs using Allegro [1]. Perhaps such an experiment is hard to run, but some evidence that the speed benefit of So3krates on MD17 translates to a large scale simulation would help the case for non-local model development.

[1] https://arxiv.org/pdf/2204.05249.pdf

**Strengths And Weaknesses:**

Overall the authors have come up with a principled approach for including long ranged geometric features into a message passing architecture. I think the method is interesting and sufficiently novel for publication, and the results that are presented are favourable but perhaps slightly incomplete (see significance section below).

Originality

The idea to perform some sort of convolution with spherical harmonic wavelets to provide additional geometric input is not new, and nor is the discussion of the equivariance of this operation. However, the interpretation of these SPHC features as living in a coordinate space where interesting neighbourhoods can be learned for message passing that correspond to very distant communication in Euclidean space is an interesting new insight that I had not thought about before.

Quality

The main takeaway for me was that the proposed model is (1) solving the specific case of cumulene highlighted in literature before (2) achieving similar performance on MD17 as a number of other methods, but with a small number of parameters and higher train/test speed. I have comments on (1) in the Questions section below and (2) in the significance section below. However I think overall the quality of results meets the publication bar because the lightweight nature of the model seems a promising direction.

Clarity

The paper is easy to follow.

Significant

I feel like the significance of the paper suffers slightly from a couple of missing baselines:

(1) I’m not an expert on ML forcefields, but I think SpookyNet makes more of a deliberate effort to model non-local behaviour than NequIP, so I was not sure why NequIP was chosen as the baseline in fig 2. A more considered description of how models like SpookyNet, PhysNet, sGDML etc capture non-locality and what sort of non-locality SPHC is particularly adept (or not!) at capturing might be more educational than the somewhat obvious (and already published) statement that a NequIP model cannot capture effects larger than it can see.

(2) I’m not sure I know how to evaluate the differences between the numbers on MD17: Although the statement that the errors are low and the parameter counts are low are true, all the models in table 1 have ~10^6 parameters in order of magnitude and all have broadly similar performance on MD17. Do these differences matter in practice? The timings of So3krates in fig 1d seem the most convincing difference to existing methods, but some of the baseline models are missing from this plot and DimeNet (the closest competitor in timing) is missing from the rest of the paper.

(3) SpookyNet has ~10-20 meV/A on the full QM7-X set, but this is not mentioned so I was not sure why So3krates was compared with sGDML rather than something like SpookyNet. Also I think I missed why So3krates is only tested on a subset of QM7-X.

I also flag a slight concern that this paper mainly addresses the cumulene problem which is not of broad interest to NeurIPS, but I personally found this interesting so I did not factor this into my score.

---

> ### Author Response · Authors · 2022-08-02
> **Authors response to the reviewer**
>
> We thank the reviewer for their helpful comments and questions regarding our manuscript, and their overall positive assessment of our work.
>
> 1) We chose the cumulene task as a representative example, because it has been introduced in earlier work as an open problem. To solve the cumulene task, a model has to be able to resolve the relative orientation of the rotors at opposite ends of the carbon chain. There are three possible ways to solve this task:
> - The model needs to be "global", i.e. no cutoff radius (or a “large enough” cutoff to “see” opposite ends), or
> - it needs to propagate local information, such that the "effective cutoff" (see Fig. 2) is large enough, or
> - it needs to include some "non-local interaction" explicitly.
> The second option only works when the model uses equivariant features (of high enough degree), because otherwise the information about relative orientation is lost. This is why we chose NequIP (a SOTA equivariant model) as baseline. We think it might not be obvious to all readers that NequIP cannot solve this task even though it can "see" far enough, at least with features of insufficient degree (l<2). However, we did not make this clear enough and additional baselines for other model classes were missing from our manuscript. We ran additional experiments with SchNet, sGDML (global), and SpookyNet (explicit non-local interaction). None of these models successfully describes the rotor-energy profile. These additional baselines and detailed per-model analyses (with explanations for their failures) will be included in the camera-ready version.
>
> 2) For many practical applications, small differences in error are indeed irrelevant, although they might be important in some cases (e.g. when spectroscopic accuracy is needed). We still chose to report the performance on MD17, because it is a commonly used benchmark and allows readers familiar with the literature to quickly gauge the quality of different models. We are thankful to the reviewer for pointing out that DimeNet is missing from Fig. 1d. We will add a discussion of DimeNet to the text, and timing results to Fig. 1d and Table 1. The camera-ready tables and figures will include the baseline models: GemNet, DimeNet, SpookyNet, PaiNN, NewtonNet and NequIP.
>
> 3) In contrast to the results reported in the SpookyNet paper, So3krates was trained individually on each structure (for only 80 data points) rather than jointly on multiple structures. As kernel models are known to be particularly strong in the low-data regime, we chose sGDML as the baseline model. Still, we agree that comparing So3krates to SpookyNet on the whole QM7-X data set is an important additional baseline. We are currently running experiments on the full QM7-X dataset. Preliminary results for a training data set of 500k structures show energy and force errors of 24 meV and 28 meV/A, respectively. We expect that training on the full training set (3.6M structures) will give results comparable to other SOTA methods like SpookyNet. The final results will be added as an additional baseline to the camera-ready version of the manuscript.
>
> Additionally, the reviewer made us aware of additional benchmarks for non-local effects presented in a paper by Ko et al.  Preliminary results on these benchmarks show that switching between the local and non-local version of So3krates helps to improve model accuracy for the carbon chain in which long-range charge transfer plays a prominent role. The final results for all benchmarks in the Ko et al. paper will be added to the camera-ready version of our manuscript.
>
> Further, the reviewer raises two points regarding (1) scaling and (2) generalization to larger molecules. To address (1), we compared the scaling behavior of
> -a partially local model (in Euclidean space),
> -of a fully local model (in Euclidean space and SPHC space) and
> -of a global model (no cutoff in either Euclidean or SPHC space).
> We observe that the model with SPHC localization scales better to large structures than a fully global model. Possible computational improvements based on this observation will be discussed in detail in the final version of the manuscript. For (2), we investigated the generalization to large unknown structures by performing geometry optimizations and comparing to ground truth relaxed structures. Preliminary results show So3krates generalizes to unknown structures much larger than those represented in the training data (e.g. a polypeptide consisting of ten amino acids). Results for these additional experiments will be added to the final version of our manuscript.
>
>
> If the reviewer feels our proposed changes to the manuscript adequately address their concerns, we kindly ask them to consider raising their score accordingly.

---

### Meta-Review · Area_Chair_wHAm · 2022-08-25

**Recommendation:** Accept
**Confidence:** Certain

**Metareview:**

The reviewers all agreed that this was a novel, interesting and effective innovation. It seems clear that, with the modifications to the paper that the authors agreed to, this paper should be accepted.

**Award:**

No

---

### Decision · Program_Chairs · 2022-09-14

Accept